# A thalamocortical pathway for fast rerouting of tactile information to occipital cortex in congenital blindness

Franziska Müller [1,8], Guiomar Niso [2,3,4,8], Soheila Samiee [2], Maurice Ptito [1,5], Sylvain Baillet [2]* & Ron Kupers [1,5,6,7]*

In congenitally blind individuals, the occipital cortex responds to various nonvisual inputs. Some animal studies raise the possibility that a subcortical pathway allows fast re-routing of tactile information to the occipital cortex, but this has not been shown in humans. Here we show using magnetoencephalography (MEG) that tactile stimulation produces occipital cortex activations, starting as early as 35 ms in congenitally blind individuals, but not in blindfolded sighted controls. Given our measured thalamic response latencies of 20 ms and a mean estimated lateral geniculate nucleus to primary visual cortex transfer time of 15 ms, we claim that this early occipital response is mediated by a direct thalamo-cortical pathway. We also observed stronger directed connectivity in the alpha band range from posterior thalamus to occipital cortex in congenitally blind participants. Our results strongly suggest the contribution of a fast thalamo-cortical pathway in the cross-modal activation of the occipital cortex in congenitally blind humans.

[1] BRAINlab, Department of Neuroscience, Panum Institute, University of Copenhagen, Copenhagen, Denmark. [2] McConnell Brain Imaging Centre, Montreal Neurological Institute, McGill University, Montreal, QC, Canada. [3] Centre for Biomedical Technology, Universidad Politécnica de Madrid, Madrid, Spain. [4] Biomedical Image Technologies, ETSI Telecomunicación, Universidad Politécnica de Madrid and CIBER-BBN, Madrid, Spain. [5] École d'Optométrie, Université de Montréal, Montréal, QC, Canada. [6] Department of Radiology & Biomedical Imaging, Yale University, New Haven, CT, USA. [7] Institute of Neuroscience, Université catholique de Louvain, Brussels, Belgium. [8] These authors contributed equally: Franziska Müller, Guiomar Niso. *email: sylvain.baillet@mcgill.ca; kupers@sund.ku.dk

A vast number of studies have shown activation of the occipital cortex in congenitally blind (CB) subjects by various nonvisual sensory inputs, such as auditory, tactile, and olfactory stimuli[1–4]. Cross-modal sensory trafficking is often attributed to cortico-cortical connectivity, yet thalamic projections may redirect sensory pathways. Congenital blindness constitutes a natural human model for investigating the pathways involved in the reallocation of the visual cortex for somatosensory processing. This condition needs to be clearly distinguished from adult post-lesional plasticity, or late-onset blindness, whereby blindness occurs after the wiring of the brain has already been firmly established. The prevailing hypothesis is that in CB subjects, auditory and tactile information reach the occipital cortex via cortico-cortical pathways that are also present in the sighted brain, but that are developmentally strengthened in cases of congenital blindness. Evidence supporting a strengthened cortico-cortical route for nonvisual afferents is derived from brain imaging studies showing stronger functional connectivity of auditory and somatosensory cortices with the occipital cortex in blind subjects[5–11]. However, these studies used functional magnetic resonance imaging (fMRI) or positron emission tomography (PET), i.e., techniques with insufficient temporal resolution to capture the fast dynamics in the millisecond range of the spatiotemporal spread of auditory or tactile information to the occipital cortex.

Rerouting at the thalamic level presents an alternate pathway by which nonvisual sensory information could reach the occipital cortex. Indeed, congenital enucleation studies in rodents suggest that the visual cortex receives input diverted from thalamic nuclei originally associated with the somatosensory and auditory cortices[12–15]. Although no data have yet been obtained to directly support this mechanism in humans, we have reported microstructural changes in the thalamus of CB subjects, which is consistent with a comparable thalamic reorganization in humans[16]. In order to disentangle the contributions of these two purported pathways of cross-modal neuroplastic responses in CB humans, we used magnetoencephalography (MEG) imaging to measure with millisecond precision the brain's response latencies to electrotactile stimulation of the finger and frequency-specific directed connectivity measures of brain rhythmic fluctuations. Stimulation was randomly applied to the left or right index finger and participants had to indicate as quickly as possible which finger had been stimulated. We reasoned that short occipital cortex response latencies would be indicative of the implication of a direct thalamo-cortical pathway, whereas longer latencies would indicate the involvement of a slower (polysynaptic) cortico-cortical pathway via the somatosensory cortex. We also hypothesized that stronger effective connectivity directed from thalamus to occipital cortex in frequency bands typically expressed in the occipital cortex (e.g., the alpha band between 8 and 12 Hz) would also be indicative of a direct thalamo-cortical pathway in CB. Despite lower signal-to-noise ratios in deep brain structures such as thalamus, recently published studies have shown that MEG imaging is nevertheless sensitive to detect activation of subcortical regions[17,18]. We therefore derived measures of regional neurophysiological activation and of directed connectivity between thalamic and cortical structures to test these hypotheses.

Our results show that tactile stimulation produces early occipital cortex activations in CB individuals, starting as early as 35 ms, and increases directed connectivity in the alpha-band range from posterior thalamus to occipital cortex. Together, these observations indicate a fast thalamo-cortical pathway in the cross-modal activation of the occipital cortex in CB humans.

## Results

**Behavioral data.** Our behavioral data indicated that for supra-threshold electrocutaneous stimuli applied to the left index finger, CB participants responded faster than the blindfolded sighted controls (SC) in detecting which hand had been stimulated (CB: 288 ± 33 ms; SC: 323 ± 31 ms; $p = 0.04$, unpaired t test; Supplementary Fig. 1). There was a nonsignificant trend for faster reactions of CB (296 ± 36 ms) compared with SC (315 ± 24 ms) in response to right index finger stimulation. No between-group differences in the percentage of correct responses for left or right index finger stimulation were found for any of the stimulus intensities used. Supra-threshold stimulation intensities were significantly lower in the blind participants for both the left (CB: 1.4 ± 0.1 mA; SC: 1.8 ± 0.3 mA; $p = 0.01$, unpaired $t$ test) and right (CB: 1.4 ± 0.1 mA; SC: 2.0 ± 0.4 mA; $p = 0.03$, unpaired $t$ test) index finger stimulation (Supplementary Fig. 1).

**MEG data.** We extracted the neurophysiological brain responses related to index finger stimulation from three anatomical regions of interest (ROIs) from full-brain distributed source modeling of the MEG data. The ROIs were defined from an atlas of Brodmann areas registered to the cortical surface of each participant (see the "Methods" section). They comprised portions of the primary somatosensory cortex (S1), primary visual cortex (V1), and the cortical surface area in contact with the posterior lateral aspect of the thalamus, i.e., including the lateral geniculate nucleus (LGN). Following left hand stimulation, early thalamic responses initiated between 10 and 25 ms were detected in both groups (Fig. 1a and Supplementary Fig. 3a). In S1, the first response peaked at 30 ± 3 ms on average across individuals, followed by a second and larger response that peaked around 50 ms (Fig. 1a). This spatiotemporal activation pattern did not differ between the blind and sighted groups. In sharp contrast, we found significantly increased activity ($p < 0.02$, two-tailed $t$ test) in the contralateral V1 region peaking as early as 35 ms after left finger stimulation (Fig. 1b) in CB subjects. This cortical response was followed by a second and larger peak around 55 ms. Overall, V1 activations were significantly stronger in the CB group compared with SC averaged between 35 and 50 ms post stimulation (Fig. 1b; $p < 0.02$, two-tailed $t$ test). A whole-brain analysis further revealed that the difference in 30–55-ms activation between groups extended contralaterally to the stimulation along the calcarine fissure and to the parieto-occipital sulcus. The respective timings of the thalamic and occipital responses to electrotactile stimulation in CB were similar to those measured in a SC subject in response to visual stimulation (Supplementary Fig. 2). The responses to right-hand index finger stimulation were also similar to these observations in thalamus and S1, and showed the 50-ms differential component in V1 between groups (Supplementary Fig. 3).

We confirmed these findings with a second analysis of the MEG data based on a linear regression model restricted to the three ROIs. We replicated the approach used by Coffey et al.[17] to further discriminate between the respective contributions from each brain region of interest to the MEG data. As explained in the Methods section, the resulting multiple linear regression model (MLR) restricts the modeling of absolute-valued sensor MEG data to the instantaneous, weighted linear combination of the absolute values of the forward fields of each ROI. The ROI forward fields were obtained from the same head model coefficients as those used for the weighted-minimum-norm estimate (wMNE) approach. The rectification of the sensor data and forward fields emphasizes the fit of each ROI's contribution to the topography of recorded MEG sensor data, regardless of the flux direction of the magnetic induction measured outside the head. The resulting linear regression coefficients obtained for each region at every

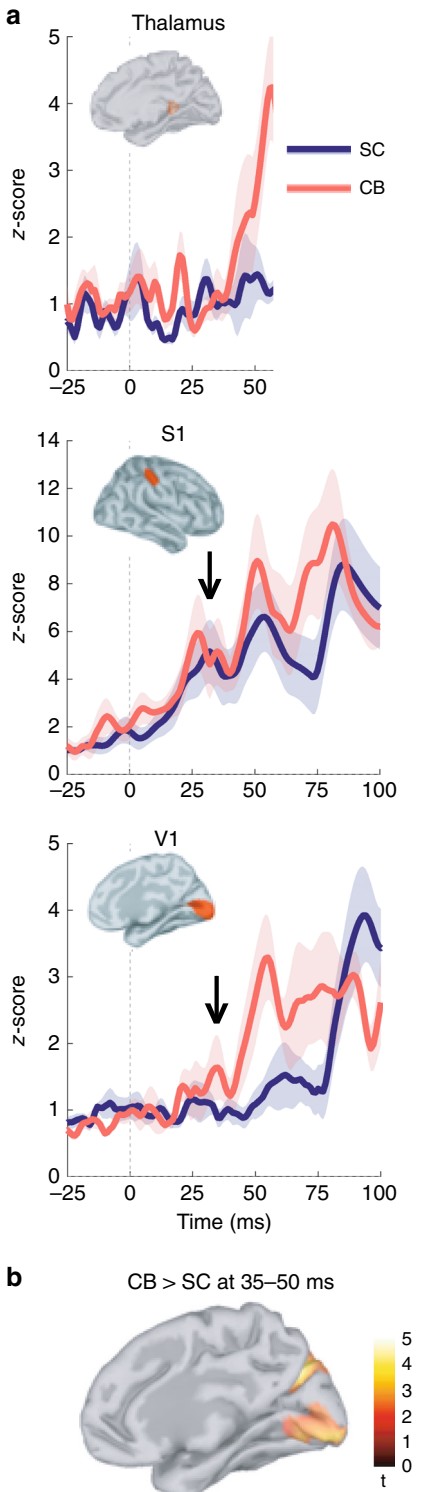

**Fig. 1** Neurophysiological responses following left index finger stimulation. **a** The source traces of MEG data were estimated across the cortical surface and extracted from regions of interest in thalamus, S1 and V1 in congenitally blind (CB) and sighted control (SC) subjects. The electrocutaneous finger stimulation occurred at 0 ms. Shading indicates the standard error of the mean (SEM) for each group. The significant S1 response in CB and SC and the early V1 response in CB only are indicated by black arrows. **b** Cortical activations between 35 and 50 ms following left index finger stimulation were stronger in CB vs. SC subjects over contralateral parieto-occipital regions ($p <$ 0.05 uncorrected, permutation test with 5000 randomizations, t-test unequal variance). Medial view of the right hemisphere

point in time therefore underline each ROI's contribution to surface data while lifting the ambiguity of estimating the direction of its current flow. This linear mixing model has enhanced spatial specificity but is strictly limited to the three tested ROIs in comparison with the distributed model of neurophysiological responses presented above (wMNE), which provided a fuller physiological account of cortical activations (i.e., by estimating both the amplitude and the direction of the current flow in all ROIs and the rest of the cortical surface, see Methods). Akin to Coffey et al.[17], who clarified cortical vs. deep and early brainstem responses to auditory stimulation by using this approach, the time series of the linear regression model coefficients showed a distinct peak around 20 ms in thalamus following finger stimulation in both groups (Fig. 2a). Similar to the distributed source data, the model did not show significant differences in thalamus and S1 between groups. It also confirmed that the early V1 response in CB participants initiated as early as 35 ms after finger stimulation, with strong differential intensity compared with SC controls until at least 50 ms poststimulus (Fig. 2b). A later contribution from V1 occurred earlier (around 80 ms) in SC than in blind participants (around 100 ms). The MLR model was based on smoothed time series over a sliding 15-ms time window centered at each point in time. The resulting temporal smoothing involved data points ±7 ms around the running data point at time $t$. For this reason, the early onset activity revealed in CB V1 at 35 ms may be contributed by regional activation over the [28, 42]-ms time window across participants. Therefore, the later responses revealed by both the MLR and the wMNE model (Figs. 1a and 2a) are not likely to have contributed to this earlier-onset V1 activation in CB, which was absent in SC. No time windowing was used in the wMNE source time series.

The effects of hemisphere and group were tested by using a mixed model ANOVA with hemisphere as within-subjects variable and group as between-subjects variable. We did not find a main effect for hemisphere ($F(1) = 1.7$, $p = 0.21$) or interaction between hemisphere and group ($F(1, 14) = 2.33$, $p = 0.15$). However, there was a main effect of groups ($F(1) = 15.6$, $p = 0.001$), reflecting stronger activity in V1 for CB subjects (Supplementary Table 4).

To evaluate possible direct communication between thalamus and V1 in blind participants, we tested for group differences in directed functional connectivity between ROIs over the epoch duration (−100 to 500 ms) around index stimulation events. We measured phase-transfer entropy (PTE) to assess directed information flow between brain regions[19] by using the ROI time series from the distributed source model of neurophysiological activity. PTE is an indicator of the strength and direction of signal interactions between narrow-band oscillatory rhythmic fluctuations of brain activity. We found significantly stronger alpha-band (8–12 Hz) connectivity directed from thalamus to V1, and from S1 to V1 in CB compared with SC (Fig. 3a). There was stronger beta-band (12–30 Hz) connectivity directed from S1 and V1 to thalamus in the SC group (Fig. 3b). We did not find significant PTE differences in the gamma-frequency range (40–80 Hz) between the tested ROIs (all tests for significance: $p < 0.05$, with Bonferroni correction).

## Discussion

To test for the contribution of a thalamo-cortical pathway in the rerouting of tactile information to the occipital cortex in CB humans, we combined MEG source imaging and directed functional connectivity to resolve the spatiotemporal dynamics of brain activation in response to tactile stimulation. In both blind and sighted participants, the earliest significant response peak was

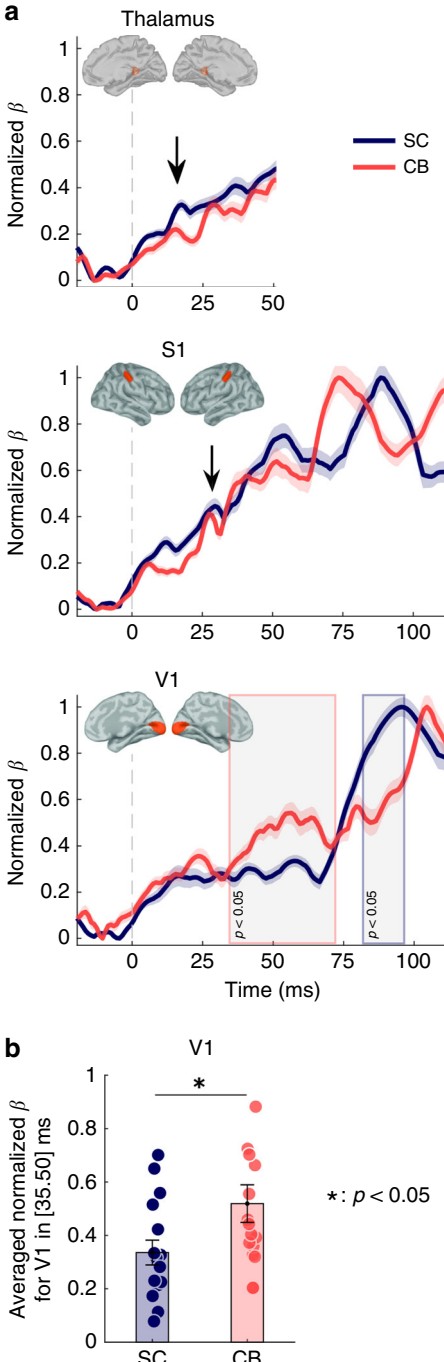

**Fig. 2** Linear regression model of ROI contributions to MEG data.
**a** Temporal variations of the linear mixing model coefficients shown in each region of interest, following electrocutaneous stimulation of the index finger (0 ms). The data from left and right index finger stimulation were pooled for greater statistical power. The linear coefficient time series are shown for contralateral posterior thalamus (top, with early peak marked with an arrow), primary somatosensory (S1, middle), and visual (V1, bottom) ROIs in CB and SC subjects. Shading indicates standard error of the mean estimates. Gray-shaded boxes highlight segments of significant differences between the two groups ($p \leq 0.05$), with their borders colored according to the group showing stronger V1 contribution (two-tailed independent $t$ test, $p \leq 0.05$—not corrected for multiple comparison due to a limited number of participants). The thalamus model coefficients are shown only over the first 50 ms after stimulus onset to emphasize the early peak around 20 ms (black arrow). **b** Average model coefficients in V1 within the [35, 50]-ms time window showed greater V1 contribution in CB vs. SC. Mixed model ANOVA with hemisphere as within-subject variable and group as between-subject variable: no main effect of hemisphere or interactions between hemisphere and group, but significant effect of group ($F(1) = 15.6$, $p = 0.001$) reflecting stronger activity in V1 for CB subjects

a significant group difference in the latency or the amplitude of the S1 response. The third measured peak was an early occipital response in CB only after around 35 ms, hence shortly after the S1 response. This early occipital response was found with both source interpretation models of the MEG data, and was followed by a second and larger occipital peak occurring around 55 ms. In the full cortically distributed source model, the early occipital peak was not significant for right-hand stimulation. We cannot rule out a lack of statistical power due to the small sample size imposed by the CB special population. We also emphasize the variability in response latencies in striate cortex. Work from Maunsell and Gibson[22] indeed showed considerable variability in striate cortex responses to visual stimuli, ranging from as early as 20 up to 70 ms. We note however that behaviorally, response times to right-hand finger stimulation were not significantly different either between blind and sighted participants.

We pooled together the data from both hand stimulation trials to fit the linear regression model. The model confirmed stronger early contribution from the occipital cortex in blind participants, starting as early as 35 ms and lasting up to 74 ms poststimulus onset. The 15-ms smoothing time window used in the linear mixing model was centered on each time point. Hence, the early 35-ms onset was contributed by V1 signals at most within the 28–42-ms time range and cannot be explained by the secondary, later activity shown by the fully distributed cortical model. Note that this latter analysis did not include moving-average temporal smoothing, and indicated an early V1 peak in the CB group around 35 ms, which was absent in SC.

We claim that the early occipital cortex response in CB participants, occurring as early as 35 ms after stimulation onset, is driven by the rerouting of tactile information to the occipital cortex via a thalamo-cortical pathway. Indeed, we measured a thalamic response latency to electrotactile stimulation around 20 ms. Since the estimated LGN-to-V1 transfer time in the human brain varies between 13 and 19 ms (average: 15 ms)[23], the earliest cortical response times for a V1 response mediated by a thalamo-cortical pathway are expected to occur after around 33–39 ms. By using graph-theoretical tools to analyze connectivity data in the macaque brain, Négyessy et al.[24] showed that the shortest paths for routing tactile information from S1 to V1 pass through the secondary somatosensory cortex (S2), to the ventral intraparietal area (VIP), from whence information is funneled to V1 via middle temporal visual area (MT), tertiary

detected around 20 ms in the posterior aspect of the thalamus by using both a full cortically distributed source model in individual subjects (showing substantial interindividual variability) and a linear regression model constrained to thalamus and the other two ROIs. This latter modeling approach confirmed its greater sensitivity and consistency for the detection of subcortical activity with MEG, as shown previously by Coffey et al.[17]. Latency and amplitude of the thalamic responses did not differ significantly between the blind and sighted participants. The observed latency of the thalamic response is in line with reported thalamic response times of 15–17 ms following stimulation of the upper extremities[20,21]. The second peak detected was in S1 contralateral to the stimulated finger and occurred around 30 ms; it was confirmed in both the full distributed cortical and the linear regression model. In line with the thalamic response, we did not detect

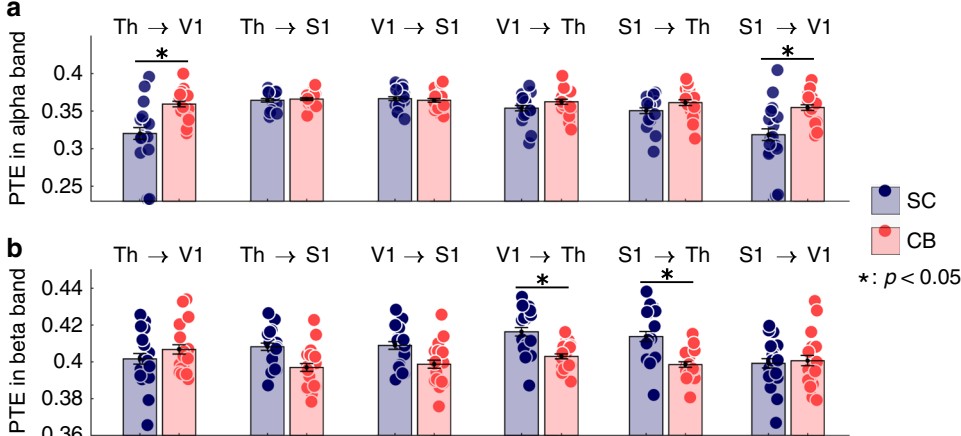

**Fig. 3** Directed functional connectivity measured with phase-transfer entropy (PTE). **a** CB subjects showed stronger directed connectivity in the alpha frequency band (8–12 Hz) from thalamus (Th) to V1 ($p = 0.019$) and from S1 to V1 ($p = 0.05$; two-tailed Wilcoxon rank-sum test, Bonferroni corrected). **b** SC subjects showed stronger PTE connectivity in the beta-band (12–30 Hz) directed from S1 ($p = 0.027$) and V1 ($p = 0.021$) to thalamus (two-tailed Wilcoxon rank-sum test, Bonferroni corrected)

visual cortex (V3), and parieto-occipital area (PO). This implies that area 3b is at least three synapses away from V1 in the primate brain. Given that the monosynaptic passage of information from one cortical area to another generally takes around 5–15 ms, it follows that an activation of V1 via a cortico-cortical pathway would result in a corresponding minimum V1 response time of around 60 ms in the human brain. This estimate fits with our previous findings showing that the timing of the V1 response following electrical stimulation of the median nerve of CB subjects occurs at 55–70 ms post stimulation[7,20]. In addition, the similar response latencies in S1 and V1 make it highly unlikely that the early occipital activation could be driven by rerouting of tactile information from S1 to V1.

The increased directed connectivity of alpha frequency band oscillatory signals from the thalamus to V1 in CB participants further supports the hypothesis of rerouting of tactile information via a thalamo-cortical pathway in this population. Alpha oscillations are prominently expressed over the posterior aspects of the occipital and parietal cortices in humans[25]. Depth recordings and multi-modal imaging studies have also emphasized the role of posterior thalamic nuclei as a generator of alpha oscillations[26]. Although occipito-parietal alpha oscillations of strong amplitude are typically associated with lower vigilance, essentially in the context of visual attention tasks, recent research has actually emphasized their role in the temporal modulation of net excitability of cellular assemblies[27]. In particular, the phase of alpha oscillations has been found to be associated with the temporal registration of perceptual items by brain systems, and in particular the visual brain[23,28]. There is evidence that alpha-band oscillations gate task-relevant visual information to downstream areas and suppress irrelevant visual information[29,30]. Our MEG findings align with this model in providing evidence of bottom-up signaling issued by thalamic oscillations towards the visual cortex in the context of cross-modal sensory stimulation. We found alpha-band directed connectivity between thalamus and V1 to be stronger in CB participants, which is consistent with the reallocation of an existing thalamo-cortical visual pathway to the processing of somatosensory inputs in the brain of CB individuals. We note that Kriegseis et al.[31] reported a reduced EEG alpha activity over parieto-occipital brain areas in CB in a somatosensory perception and mental imagery task. This finding is not in contradiction with our data, which report effects in alpha interregional directed connectivity rather than differences in

power. Further, we report occipital activity from the primary visual cortex rather than from an extended parieto-occipital region.

We further reported stronger alpha-band directed connectivity from S1 to V1 in the CB group, which is also compatible with the reallocation of occipital regions to somatosensory processing in the blind. In contrast, control subjects showed stronger directed connectivity from V1 and S1 to thalamus in the beta- (12–30 Hz) frequency range. Recent empirical evidence has shown that task-related beta-oscillations are likely to convey top–down signaling across brain networks and systems[18,32,33]. Our data therefore point at possibly more pronounced feedback processes in sighted participants in the context of somatosensory stimulation, which may be attributed to the requirement of augmented sensory processing resources in the unfamiliar blindfolded conditions.

Although it is typically assumed that oscillations in the gamma-frequency range subserve cortico-cortical, bottom-up communication, we did not find significant differential effects of gamma-directed connectivity between the experimental groups. It is possible that the PTE measure is challenged by the lower signal-to-noise ratio at higher-frequency ranges, especially in deeper brain structures, making the estimation of phase angles of low-amplitude signals less sensitive[18].

Previous studies did not report increased connectivity between the thalamus and the occipital cortex in congenital blindness[8,20]. The technique we used for measuring directed functional connectivity (PTE) is designed to be more specific of narrow-band and fast rhythmic fluctuations of brain signals, compared with dynamic causal modeling of the BOLD fMRI signal[8] or time-delayed mutual information of MEG data[20]. Both of those methods have limited temporal resolution and are not sensitive to directed transfers between narrow-band oscillatory brain signals. This may have limited their ability to identify fast-coupling phenomena between subcortical and cortical brain regions.

We observed a second and larger activation peak in the occipital cortex of the CB participants at a later latency (55 ms) and sustained increased activity with a poststimulus latency up to 100 ms. The timing of this later activity is compatible with a multisynaptic cortico-cortical pathway conveying tactile information to the occipital cortex. Together, these data suggest that occipital activation by nonvisual (i.e., tactile) input comprises at least two components, one subserving fast thalamo-cortical signaling and the other slower cortico-cortical signaling.

We also detected a strong activation in V1 of blindfolded SC subjects that peaked around 95 ms. fMRI studies have previously described primary visual cortex activation in sighted individuals in a tactile discrimination task[34–37]. Using transcranial magnetic stimulation (TMS) to inhibit occipital cortex interfered with behavioral performance in a tactile orientation discrimination task in normally sighted individuals[38].

Previous studies of cross-modal neuronal plasticity in humans have argued that nonvisual information is conveyed to the occipital areas via a cortico-cortical pathway[5–8,20]. The related unmasking hypothesis proposes that loss of a sensory input enables unmasking and strengthening of preexisting neuronal connections between heteromodal sensory cortical areas. The unmasking hypothesis further posits that under normal circumstances, these preexisting connections mediate sensory integration, such that tactile and auditory information conveyed to the occipital cortex normally enable visual processing[39], while being insufficient to drive visual sensations in the presence of a normal visual input. In this scenario, the cortico-cortical pathways are strengthened and become functionally unmasked following visual deprivation. Our results confirm the expected strengthening of the cortico-cortical pathway in CB individuals and suggest that the phase of alpha-band oscillations mediates this strengthening. We also provide the first empirical evidence that cortico-cortical cross-modal signaling is accompanied by a faster thalamo-cortical functional connection. This raises the question of the relative contributions of the thalamo-cortical and cortico-cortical pathways in cross-modal plasticity. Phrased differently, would the thalamo-cortical pathway be sufficient to explain by itself cross-modal responses? Although we cannot provide a conclusive answer to this question based solely on the present findings, it is unlikely that thalamo-cortical connection would be sufficient to fully explain cross-modal plasticity. First, the magnitude of the thalamo-cortical response is quite moderate compared with that of the later cortico-cortical response. Second, there is evidence that blocking the cortico-cortical pathway by applying TMS over S1 interferes with cross-modal plastic responses[5]. We suggest that the arrangement of a dual pathway to the occipital cortex in the CB brain allows early thalamo-cortical signaling to modulate ensuing occipital responses communicated by the slower cortico-cortical pathways, facilitating reallocation of the visual cortex to tactile processing.

Evidence of brain rewiring from animal studies indicates that tactile and/or auditory information can be rerouted to the visual cortex via altered thalamo-cortical connections[13–15]. After applying retrograde tracer injections in visual cortical area 17, Karlen et al.[14] observed labeled cells in the somatosensory thalamus of the binocularly enucleated opossum[14]. Taken together, these animal studies suggest that the congenital absence of vision from birth is permissive to rewiring at the thalamic level. We argue that such subcortical rewiring also occurs in CB humans; in this scenario, cross-modal responses in the occipital cortex would be mediated by fibers of the auditory or somatosensory thalamic nuclei invading the denervated LGN[40]. Although reduced in size in blind individuals, the LGN and the optic radiations relaying information to V1 are anatomically present[41–43], suggesting that they may continue to undertake some function, despite primary denervation. Since the LGN does not receive retinal inputs in CB individuals, the integrity of its connections must be sustained by nonvisual afferent thalamic or cortical inputs[41]. We propose that tactile information is rerouted through alternate connections from the somatosensory thalamus (VPL) to area V1 via the LGN. This model is also supported by animal studies showing increased fiber density originating from the lateral posterior and the somatosensory ventroposterior nuclei of the thalamus to the LGN in congenitally anophthalmic mice[12], and by LGN activation

upon auditory stimulation in the blind mole rat[11]. Results from other animal studies by using a different model of central rewiring of visual afferents are in line with the proposal that cross-modal activity may be mediated by plastic changes in subcortical pathways[44,45]. Previous work from our laboratory by using diffusion tensor imaging revealed decreased fractional anisotropy in several clusters within the thalamus of CB subjects, suggesting a change in the microstructural distribution of fiber connections within the thalamus, possibly related to altered internal connectivity between the clusters[16]. Our present functional connectivity results between the thalamus and V1 further underscore the hypothesis of thalamic rewiring.

Figure 4 shows a schematic depiction of our proposed dual-pathway model of the rerouting of nonvisual information to V1. The biological significance of the thalamo-cortical pathway lies in the rapid transfer of somatosensory information to the occipital cortex. This arrangement also suggests that the early thalamo-cortical-mediated activity may modulate ensuing occipital responses induced by slower cortico-cortical pathways, thus facilitating top-down reallocation of the visual cortex to tactile processing. Behaviorally, this provides mechanistic support to the observation of faster reaction times to tactile inputs in our CB participants. In their daily life, blind individuals may benefit from such functional adaptation, for instance in the fast processing of rich tactile information such as in Braille reading.

We are aware of some study limitations. The present evidence for redirection of tactile information to the CB visual cortex is based on response latencies of occipital activations, and on increases in functional connectivity between electrophysiological signals in thalamus and V1. We reasoned that if tactile information is conveyed to the occipital cortex via a cortico-cortical pathway, the earliest measurable activation would not occur before 60 ms. Although it is theoretically possible that tactile information could be relayed directly from S1 to V1, as between the secondary auditory cortex and V1 in the macaque brain[46], we are unaware of studies indicating a direct pathway from S1 to V1 in the human brain. A more direct proof for the existence of thalamic rewiring could in principle be achieved by high-resolution diffusion imaging[16], by postmortem studies of the brains of CB subjects, or by fluorodeoxyglucose PET (FDG-PET) studies focusing on the LGN after tactile stimulation. Alternatively, combined TMS of S1 with recording of neural responses in the occipital cortex in response to tactile stimulation might also provide a more direct demonstration of occipital responses relayed through a thalamo-cortical pathway.

Since all our CB participants were Braille readers, another limiting factor is that some of the observed changes may not relate to blindness per se but to the fact that our subjects rely more heavily on tactile input. There is indeed evidence from the literature that increased reliance on the sense of touch is the trigger for both behavioral and brain functional and structural changes[47–49].

Although there is growing evidence that sources can be assigned to deeper brain regions with MEG source imaging[15], the associated low signal-to-noise ratio is a challenge for signal extraction[18]. PTE is one among the possible measures of functional connectivity in electrophysiology, which offers a measure of anatomically directed interactions between narrow-band oscillatory neural signals. As such, PTE is more directly related to the neurophysiology of MEG source signals and avoids the pitfalls of large-band parametric statistical modeling of effective connectivity[50].

In summary, functional source imaging of MEG data revealed that V1 in CB subjects is activated as early as 35 ms after tactile stimulation, in concert with an S1 activation. This suggests a rerouting of tactile input at the thalamic level, possibly through a

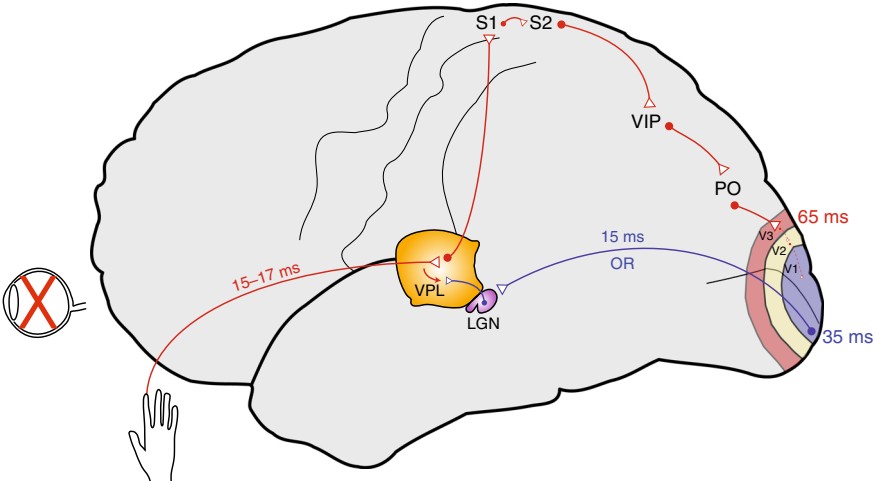

**Fig. 4** Proposed scheme showing tactile information flow to the occipital cortex. Tactile information from large myelinated (Aβ) afferents is sent to the VPL and then rerouted to the LGN, from which it is further relayed to the primary visual cortex via the optic radiations. This de novo pathway (in blue) allows fast transmission of tactile information to the occipital cortex in congenitally blind subjects, as witnessed by short occipital response times around 35 ms. Tactile information can also be rerouted via a strengthening of an existing cortico-cortical pathway (in red). Tactile information is relayed from the VPL to SI, and from there further to the occipital cortex via the posterior parietal cortex. Because it involves more synapses, this is a much slower pathway, with response times around 65 ms. *Abbreviations*: LGN lateral geniculate nucleus, OR optic radiations, PO parietal–occipital area, S1/S2 primary/secondary somatosensory cortex, VIP ventral intraparietal area, VPL ventroposterolateral thalamic nucleus, V1 primary visual cortex, V2/V3 visual areas V2 and V3

redirection of fibers originating from the somatosensory thalamic nuclei to the LGN. Analysis of interregional effective connectivity supports this model, showing increased directed influence from thalamic to occipital areas in CB in the (alpha) frequency band of neural oscillations traditionally linked to temporal modulation of regional excitability and the registration of sensory inputs. Our study provides empirical evidence of a thalamo-cortical pathway involved in the cross-modal activation of the occipital cortex in congenital blindness.

## Methods

**Participants**. We included eight CB (3 females, 41.4 ± 14.1 years) and eight normal-sighted control SC subjects (SC, 4 females, 40.4 ± 14.8 years) in the study. Demographics of the blind participants are described in Supplementary Table 1. None of the blind participants reported any residual vision. SC subjects were blindfolded just before the start of the MEG measurements and remained blindfolded until the end of the study. None of the participants had recent episodes of psychological or neurological diseases and none of them took psychotropic or other drugs of abuse. All participants provided written, informed consent. The study was approved by the local ethics committee of the University of Copenhagen (Region Hovestaden; Protocol nr: H-6-2013-004) and was conducted in accordance with the Declaration of Helsinki. Participants received the equivalent of 80 USD upon completion of the study.

**Stimulation**. Electrocutaneous stimulation of tactile afferents was administered with bipolar ring electrodes placed around the left and right index fingertips. Because of its sharp stimulus onset, electrocutaneous stimulation allows highly precise time calibration of the MEG signal. Prior to the MEG experiments, stimulation intensities were assessed for each subject by using an adaptive staircase method. We first determined the threshold (THR) by using a 1-step-up, 2-step-down procedure (7 reversals; minimum number of trials = 30), which was used to define the minimal stimulus intensity at which participants detected 70% of the stimuli. Next, we determined the comfort limit (CL) by using a 3-step-up, 1-step-down procedure (7 reversals; minimum number of trials = 30), to define the stimulus intensity that was 80% of the time described as perceptually very distinct, without being painful. In accordance with previous studies by using electrocutaneous stimulation[51], we calculated the supra-threshold (SUPRA) intensity as SUPRA = CL + (0, 2*r), in which r = CL−THR. Pilot studies in a separate group of normal-sighted subjects confirmed that the SUPRA intensity chosen using this method is perceptually distinct, non-painful, and does not perceptually mask the intermixed stimuli at THR intensity. The staircase and the stimulation paradigm were programmed with PsychoPy, a freely available software for psychophysics[52,53]. The scripts were run by a PC, which controlled the timing of events (sounds and electrical stimuli) and the unique trigger associated with each event. Triggers were sent via the parallel port to another PC that was used for data

recording, and were sampled simultaneously with the MEG recordings. Auditory cues (simple tones, duration = 50 ms; trigger delay = 14 ms) were produced by means of a custom-designed sound device (KAR Audio, Finland) that was located outside the shielded room. The generated sounds were conveyed to participants via plastic tubes connected to ER-14A ear tips (Etymotic Research Inc., USA). We used two stimulus current generators (SGC V3.0, SN, DeMeTec, Germany) to generate single electric square pulses (200 μs), which were delivered via the digital ring electrodes to participants' left and right index fingers (trigger delay = 70 μs).

**Task**. We used a simple reaction time paradigm and three different trial types (normal, trial-cue only, and non-trial-cue only; Supplementary Table 2). For the "normal trials", we used a 2 × 2 × 2 design with two stimulation sides (left and right), two stimulus intensities (THR and SUPRA), and two attentional conditions (attended and non-attended). Each trial began with an auditory cue that was presented to either the left of the right ear, calling on participants to focus their attention on their left or right index finger, respectively. Following a time interval ranging between 1000 and 1500 ms, an electrotactile stimulus, either at THR or SUPRA intensity, was delivered to the left or right index finger. In 75% of these trials, the side of the auditory cue correctly signaled the side of the ensuing tactile stimulus (attended condition), whereas it was incongruent in the remaining 25% of the trials (unattended condition). The subjects were instructed to indicate as quickly as possible, by pressing one of two response buttons, the hand that had been stimulated. We used nonmagnetic fiber-optic response pads with the 932 interface (Cambridge Research Systems, Rochester, UK). Signals from the response buttons were logged on the stimulation PC. An auditory cue indicated the end of the trial. The "trial-cue only" trials started with the auditory cue presented to the left or right ear, but were not followed by tactile stimulation. A trial-end cue indicated the end of the trial. During the "non-trial-cue only" trials, a different binaural auditory cue was presented, indicating to the participants that no electrotactile stimulation would be presented. Subjects were instructed not to react to this kind of trial. Again, the trial-end cue indicated the end of the trial. Here we only show the results from the attended supra-threshold stimulation condition. The results of the "trial-cue only" and "non-trial-cue only" trials are not reported in this paper.

The minimum number of trials per condition was set to 40. Hence, the attended and unattended conditions of a normal trial type were repeated 120 and 40 times, respectively, for each of the two stimulus intensities. Both the "trial-cue only" and "non-trials-cue only" trials were repeated 40 times each. Together, this comprises a total of 720 trials per experiment (Supplementary Table 3). The experiment was subdivided into 10 sessions of 72 trials each. As each trial lasted for 3 s, with a jittered intertrial interval between 1.5 and 2 s, one session lasted ~6 min. Short intersession breaks of around 1–2 min allowed participants to relax and adjust body posture.

**Magnetoencephalography (MEG)**. MEG data were acquired at the Center for Functionally Integrated Neuroscience (CFIN), Aarhus University Hospital, by using a 306-channel Elekta Neuromag TRIUX system (Elekta Oy, Helsinki,

Finland) arranged in 102 triplets of two orthogonal planar gradiometers and one magnetometer (sampling rate 1 kHz, online band-pass filter 0.10–330 Hz). The MEG was installed in a magnetically shielded room with active flux compensation (MaxShield technology, Elekta). Head position was measured at the beginning of each session by using four head position indicator coils (HPI) lightly affixed to each participant's head (two on the forehead and one on the left and right mastoid). The position of the HPI coils, anatomical landmarks (LPA, RPA, and Nasion), and the subject's head shape were digitized by using a 3D Digitizer pen (Polhemus Fast-track). Simultaneous electrooculogram (EOG) and electrocardiogram (ECG) recordings were also acquired, for detection and subsequently corrected of MEG traces for ocular and cardiac electrophysiological artifacts. All participants sat in an upright position with eyes closed during the recording sessions.

**Magnetic resonance imaging (MRI)**. For source modeling of MEG signals, structural MRI images with 1-mm isotropic voxels were obtained by using a 3D magnetization-prepared rapid gradient echo (MPRAGE) sequence at 3 T, on either a Tim Trio (TR = 2420 ms, TE = 3.7 ms, flip angle = 9°, and inversion time = 960 ms) or a Skyra (TR = 2300 ms, TE = 3.8 ms, flip angle = 8°, and inversion time = 938 ms) scanner (both by Siemens Medical Systems, Erlangen, Germany).

**Data analysis**. Data preprocessing started with removal of environmental noise from MEG signals with MaxFilter software with default parameters[54] (version 2.0 ElektaNeuromag). Notch filters were applied to remove power line artifacts around 50 Hz and harmonics. Cardiac and eye movement artifacts detected in the ECG and EOG signals were corrected by using Signal-Space Projection tools in Brainstorm[55]. An empty-room noise recording was also collected prior to each acquisition session, to capture environmental noise conditions for subsequent offline data preprocessing and source modeling by using empirical noise statistics. The empty-room MEG data were preprocessed in the same manner as the task data.

For source reconstruction, participants' individual scalp and cortical surfaces were segmented from MRI volume data by using Freesurfer[56], with default parameter settings. For every run and session, the estimation of the head position under the MEG helmet was refined by using the digitized head points. MEG forward and inverse modeling steps for source reconstruction were subsequently completed by using Brainstorm[55] with multi-sphere analytical approximation for head modeling[57] and wMNE with unconstrained source orientation for source imaging. We used the default Brainstorm parameters: depth-weighting order of 0.5, noise-covariance regularization of 0.1, and minimum-norm regularization hyperparameter of 3[58].

For the group analysis, MEG data were bandpass-filtered between 5 and 80 Hz by using even-order, linear-phase finite-impulse-response filters (also in Brainstorm) and subsequently epoched between [−1750 and 500] ms with respect to the delivery of the somatosensory stimulus. About 120 trials were averaged per condition, for every participant. The norm of source time series at each voxel location was converted to Z-score with respect to a baseline taken immediately prior to the auditory stimulation [−1750, −1500] ms. Individual source data were then projected to a Colin27 brain template[59], and spatially smoothed[60]. Group differences for the 30–55-ms time window were computed for S1, V1, and Thal, by using a two-tailed t test. A nonparametric permutation t test[61] was used to assess vertex-by-vertex brain differences between the groups (Fig. 1b).

Next, we used a linear regression model of ROI contributions. Thereto, the coefficients reflecting the linear contribution of each tested ROI were estimated following the approach proposed by Coffey et al.[17]. The sensor topography of each ROI was used as a set of regressors to model the observed event-related MEG sensor data, by using a series of consecutive linear regression models over the epoch duration.

The methodological details consist of the following procedure: first, by using the forward field of the sources within each ROI, we obtained the topographical signature of each ROI over the MEG sensor array; second, each ROI sensor topography was rectified and scaled between 0 and 1 to reduce the influence of regional depth and size; third, the scaled topographies were used as regressors in multiple linear regression over the epoch length, and the observations were set to be the measured MEG event-related responses at each trial. Right- and left hand-finger stimulation trials were modeled by using the ROIs contralateral to the stimulation (thalamus, S1, and V1). The time series of beta-coefficients of the linear model were then scaled between 0 and 1, for standardization across participants. For the same reason, the mean baseline beta-value in each ROI was removed in each participant. The resulting coefficient values were averaged between the right and left homologous ROI in each group. Finally, the averaged coefficients were normalized across time between 0 and 1 for each ROI and group.

Directed functional brain connectivity (or *effective* connectivity) was assessed by using PTE[19] of the MEG source time series between the thalamus, V1, and S1 along an extended time window (−100 to 500) ms around the tactile stimulation event. The first component extracted with principal component analysis (PCA) across sources in each region of interest was used for PTE calculation on a single-trial basis. All trials for each subject were concatenated for PTE computation. PTE was extracted in the alpha- (8–12 Hz), beta- (12–30 Hz), and gamma- (40–80 Hz) frequency bands. Group differences were assessed by using a two-tailed Wilcoxon rank-sum test, with Bonferroni correction for multiple comparisons across region pairs and frequency bands tested.

**Reporting summary**. Further information on research design is available in the Nature Research Reporting Summary linked to this article.

## Data availability
The data that support the findings of this study are available from the corresponding authors upon reasonable request.

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

## Acknowledgements
We thank Dr. Christopher Bailey and members of the Centre for Functional Integrative Neuroscience (CFIN) at Aarhus University, Denmark, for expert technical help during the MEG and MRI data acquisition, Dr. Andreas Ioannides for methodological discussions at the onset of the study, and Prof. Paul Cumming for critical reading of the paper. S.B. was supported by a Discovery Grant from the National Science and Engineering Research Council of Canada (436355-13), the NIH (1R01EB026299-01), a Tier-1 Canada Research Chair in Neural Dynamics of Brain Systems, and a Platform Support Grant from the Brain Canada Foundation (PSG15-3755). R.K. was supported by a grant from the Lundbeck foundation, Denmark. G.N. received financial support from the AXA Research Fund. M.P. was supported by the Harland Sanders Chair in Visual Science (Université de Montréal).

## Author contributions
R.K., F.M., and M.P. conceived and designed the study; F.M and R.K. performed all experiments and collected data; F.M., G.N., S.S., S.B., M.P. and R.K. analyzed data; R.K. and F.M. wrote the original draft; F.M., G.N., S.B., M.P. and R.K. wrote, reviewed, and edited the final version.

## Competing interests
The authors declare no competing interests.
