## [Peer Review File · Nature Communications]

Reviewers' comments:

Reviewer #1 (Remarks to the Author):

The authors present novel electrophysiological evidence on thalamocortical rerouting in congenitally blind subjects related to tactile processing. The methods used in the study are relevant and wisely selected. Statistical approaches used are valid. The manuscript is well written and easy to follow.

The manuscript has some minor issues which are listed below.

Blindfolded CS is stated in the manuscript that the methods part does not describe how the CS subjects were blindfolded.

The way how range is expressed in the manuscript varies, e.g., 15-17 ms and 5 – 15 ms. The authors should use one common way. I personally would prefer 15–17 ms format.

Page 2, Abstract:

The authors use the term "blind brain" (abstract and discussion page 6). I guess that it would be better to use a term "brain in congenitally blind subjects" or similar.

Line 4 MEG => magnetoencephalography (MEG)

Line 7 The abbreviation LGN has not been defined.

The abbreviation V1 has not been defined.

Line 10 blind participants => congenitally blind subjects

Page 3, Introduction:

Line 22 The abbreviation MEG has not been defined.

Page 4, Results Behavioural data:

Line 1 The authors should indicate what kind of stimuli were used and where they were applied in the first sentence of the results section. Now the first sentence is unclear.

A suggestion: The behavioural results indicate that supra-threshold electrocutaneous stimulation applied on the left index finger resulted in faster reaction times in CB compared with SC subjects.

How about the right index finger stimulation? Any statistics available?

Line 2 The abbreviation SC has not been defined.

Page 4, Results MEG data:

Line 1 The abbreviation SI has not been defined.

Line 7 Clumsy sentence: "In CB (but not in blindfolded sighted) participants, we identified significantly increased activity above pre-stimulus baseline ($p < 0.019$) in the contralateral occipital cortex as early as 35 ms following left index finger stimulation (Fig. 1b)."

Rephrase it, e.g., "We identified significantly increased activity above pre-stimulus baseline ($p < 0.019$) in the contralateral occipital cortex as early as 35 ms following left index finger stimulation (Fig. 1b) only in CB subjects."

Page 5

Line 4 The abbreviation V1 has not been defined.

Line 13 The abbreviation FDR has not been defined.

Page 6

Line 1 The abbreviation SII has not been defined.

Line 2 The abbreviations MT, V3, and PO have not been defined.

Line 13 "Alpha oscillations are prominently expressed over the posterior aspects of the occipital and parietal cortices in humans.²³ Is the citation correct?

Line 25 "... blind brain." See the comment above (Page 2 abstract)

Page 7

Line 2 beta frequency range has not been defined.

Line 16 The abbreviation PTE has not been defined.

Page 10

Line 16 FDG-PET has not been defined.

Page 13

Line 8 the brand and make of the response pads is missing.

Page 14

Line 9 Polhemus Fastrack – country of the manufacturer is missing.

Page 24

Figure 2 TH, V1, S1 have not been defined in the figure nor in the figure legend.

Control and Blind are used although CS and CB are being used elsewhere

References:

#23 is the citation correct in the manuscript? It does not state anything about parietal alpha activity.

#17 and #18 have different format with the year compared with the others.

Figure 1. The colors indicating standard deviations are not clearly visible, especially the yellow shade.

Supplementary material

Supplementary Figure 2. Since there are no space limitations here, I would prefer to see the responses from thalamic region in each CB and CS subjects.

Supplementary Figure 3. How many averages were used for event-related time series?

Reviewer #2 (Remarks to the Author):

This manuscript proposes that congenital blindness leads to cross modal reorganization of thalamic projections to V1 relaying somatosensory information to primary visual cortex. The conclusion is based on MEG data from 8 congenitally blind (CB) and 8 sighted controls (SC) in terms of source activity in time (relative to the sensory event) and brain anatomy. There is also functional connectivity data assessing strength of functional coupling across Thalamus, V1 and S1.

The data contribute to an important and longstanding question regarding the where and how of cross modal reorganization after visual sensory deprivation in humans, the conclusions are novel and of relevance to the wider scientific community.

As the authors summarize appropriately in their introduction to date work in human participants supports re-organization along cortico-cortical routes, whilst not directly supporting or ruling out reorganization via routes such as thalamus due to limitations in the method. The MEG method used here has high temporal resolution and ability (via source reconstruction) to link activity to

certain neural structures even deeper in the brain, and both of these features make it a suitable tool to get at this question in humans. It is my opinion that such a finding is important and would influence thinking in the field.

Yet, in its present form it is my opinion that the data as reported do not support the conclusions drawn. My concerns can be addressed through clarification in writing and by a suitable control experiment or additional data analysis.

My major concerns are (1) lack of reporting of relevant results in response to right finger stimulation (2) lack of sufficient separation of early responses (possibly thalamic projection driven) from later responses (possibly cortically projection driven) in V1 response average and spatial maps (3) lack of a visual control task in sighted control subjects (4) lack of reporting of group data for thalamic responses.

I detail my concerns below. I also have other, more minor, suggestions, which are outlined separately.

Major concerns

(1) Lack of reporting of relevant results in response to right finger stimulation. Reading through the manuscript it felt as if all results were only reported for left finger stimulation, but not right finger. I was not sure if no result was found for the right finger or if it was just not reported? If the results did not replicate with the right finger it is a reason for concern, unless you could explain why thalamic projections would reorganize differently for the two hands.

(2) lack of sufficient separation of early responses (possibly thalamic projection driven) from later responses (possibly cortically projection driven) in V1 response average and spatial maps. The V1 responses driven by thalamic projections are expected to occur around 35ms latency. In fact, in the discussion you outline that the late component around 55ms is more likely to reflect contribution of cortical projections to V1. Yet, for spatial maps (Figure 1d) and averages (Figure 1c) you average from 30 to 55ms. Looking at Figure 1b suggests that this will then include the second peak at ~55ms, which seems separate from the first peak at about 35ms, and which is certainly too late to provide strong support for thalamic projections. As a consequence I am not sure that the spatial maps and averages you show are indeed linked to the early response as you state, or if they rather reflect the later responses in which case they are not supporting thalamic projection changes. I suggest redoing these analyses using instead an integration window of 20-35ms or 20-40ms. It might also be useful to compute averages and maps sequentially for 20 or 25ms bins shifting them by 1ms each and visualize them as they change over time. If there is indeed a thalamic projection averages in V1 and spatial maps w/r to V1 should diverge between blind and sighted from 30-35ms.

(3) Lack of a visual control task in sighted control subjects that would strengthen argument. As I perceive it, the main support in favour of the conclusion that there is cross-modal reorganization in thalamic projections towards V1 is based on the speed of the V1 response in blind participants (Figure 1b) and differences in spatial distribution of responses between blind and sighted participants (Figure 1c). W/r to those see also my previous comment. Whilst I appreciate the data as shown w/r to S1 (Figure 1a) showing that the two groups do not differ in terms of time courses in S1, using the same methods etc with a visual response in V1 in sighted controls would strengthen the conclusion that CB participants do indeed use 'visual' thalamic projections for relaying somatosensory information to V1. This is also the case because standard deviations around the mean in V1 at that early time point (35ms) are large and overlap between CB and SC participants (as far as I can tell based on Figure 1b), thus visual control data would show how this compares to visual response in SC. With respect to the functional connectivity analysis (PTE) I was surprised about the results in the beta band, where sighted show more connectivity from V1 and S1 to Thalamus as compared to blind. Whilst I appreciate the interpretation of this finding in the discussion, it would be good to demonstrate that the method will recover coupling in a visual task

in sighted controls. This would strengthen the argument.

(4) Lack of reporting of group data for thalamic Responses. Two examples of thalamic responses are provided in the supplemental materials. Yet, it is my opinion that if you argue that your method reliably measures thalamic responses (as needed for your PTE analysis) I strongly suggest including group data for those responses.

Minor concerns

(5) lack of consideration of effects of Braille experience/reliance on touch vs. blindness. You do not report details on participants, incl. usage of Braille or touch. There is the possibility that if all blind participants are proficient and regular users of Braille the reorganization is in response to prolonged use of tactile information and not blindness. These aspects of blindness (i.e. lack of visual input vs stronger reliance on other modalities such as touch more generally or Braille specifically) are difficult to disentangle. You also show that the people who are blind in your study had lower threshold levels, suggesting heightened sensitivity for somatosensory stimulation. Thus it would be good to discuss this aspect w/r to your interpretation.

(6) Lack of detail of blind participant characteristics, i.e. level of remaining vision/ light perception, use of Braille. See also my comment (5)

(7) Figure 1 c and d have been reversed with respect to the caption.

(8) Previous work has shown differences in alpha band activity in blind compared to sighted people (Kriegseis et al. Reduced EEG alpha activity over parieto-occipital brain areas in congenitally blind adults. *Clinical neurophysiology* 117, no. 7 (2006): 1560-1573) which might be worth mentioning in the context of your results re connectivity in alpha bands.

(9) It was not entirely clear to me which trials were entered into the analyses. You have Comfort level and supra level trials and attended and non-attended. Which trials were used or were they all combined? Please clarify.

(10) Please specify which nuclei your ROI 'posterior thalamus' contains.

(11) I was wondering if you had also considered contributions and cross modal reorganizations from Superior (or possibly even inferior) colliculus and relevance to the responses you measured in V1. Animal work suggests that these nuclei and their projections show structural and functional changes in response to sensory/vision loss.

Reviewer #3 (Remarks to the Author):

General comments regarding the manuscript:

Are they novel and will they be of interest to others in the community and the wider field? Yes, they are of general interest to a wider field

Is the work convincing? Yes

On a more subjective note, do you feel that the paper will influence thinking in the field?

Absolutely

We would also be grateful if you could comment on the appropriateness and validity of any statistical analysis, as well the ability of a researcher to reproduce the work, given the level of

detail provided: Yes, statistics are appropriate and the work can be reproduced by others

Specific comments regarding the manuscript:

People with blindness since birth (congenital blindness, CB) have particularly well developed other senses like hearing and touch. The biological basis of these "super-normal" abilities is an altered brain developmental organization (not re-organization) whereby cortical regions normally used for visual processing (visual cortex) are "invaded" by neuronal connections from the other sensory system. It is generally thought that this "cross-modal plasticity" involves the alteration of cortical connectivity, whereby, for example, tactile stimulation is rerouted from the primary somatosensory cortex via different cortical connections to the occipital cortical regions where (in normal subjects) vision would have resided.

The authors have now shown for the first time that such cross-modal plasticity involves a much more fundamental brain connectivity change whereby the reorganization is already taking place in the subcortical (thalamic) region, i.e. the first relay nucleus where the different sensory systems meet. In another words, it is this fundamental first step that could explain cross-modal plasticity, long before information arrives at the cortical level. According to their paradigmatically new proposal, it is the short VPL-to-LGN reorganization which could explain cross-modal plasticity. The key experimental observation by the authors was that occipital cortical activation at 35 ms following tactile stimulation of the index finger using MEG recordings. In contrast to previous PET and fMRI studies, this technique had an excellent temporal resolution which allowed the measurement of such fast changes.

This new discovery is of significant public interest as it not only informs us of the biological basis of the enormous brain plasticity potential of the human brain but in that it provides a convincing explanation of a long-sought answer as to why people with early vision loss have improved ("super-normal") performance in other sensory system.

The study was well designed, the techniques used are up-to-date and the data analysis appropriate, the the conclusions are well grounded in the data.

There is no doubt that this paper is a highlight in the field of cross-modal plasticity and deserves – following minor revision – publication in Nature Communications. It is a discovery that will draw attention not only by specialists in the field but also by the public at large

Minor comments worth considering for revising the paper.

Page 2: Abstract

The first sentence should include the info "...congenital blindness" or "...human brain blind since birth" or similar.

The authors should state also clearly, that they looked at a case of developmental plasticity, i.e. a modification of brain organization so that readers do not confuse this with adult post-lesion plasticity or reorganization (this comment also applies to the introduction and discussion)

The abstract does not do justice to the fundamental discovery of the work. The rationale for this experiment as stated is that "...the contribution of a sub-cortical pathway cannot be excluded". The argument that something can not be excluded is not an argument that raises excitement – it is underselling the significance of the paper. The "helicopter" description above may help the authors to come up with a better (more substantive and sophisticated) argument why the study was carried out and why it solved a milestone problem of brain developmental plasticity.

Add "...blindfolded sighted control subjects"...

P 3 Introduction: Again, the authors should point out that developmental plasticity is different from adult post-lesion plasticity. In other words, there may be many similarities between both, but during development fiber connections are just beginning to be formed (organization), whereas in adulthood lesions/damage, they are already in place. Thus, while such studies with congenitally blind can give us hints for any conclusions about plasticity of the human brain in adulthood, developmental plasticity is a different matter. That is a limitation which deserves to be mentioned

P4 Results: To facilitate readability, the first sentence in this section should mention that the authors compared CB participants with normally sighted subjects which were blindfolded for xxx

min/hrs.

P5 Discussion: The discussion should start off with an short "helicopter" view of the study rationale, then state the key finding again

P8: Last sentence of 2nd paragraph (... "Our results also provide..."). Here it would be helpful if the authors state if the thalamic reorganization coexists as a "second" mechanism of transmodal plasticity or if it "replaces" the concept of cortico-cortical reorganization. In other words, is it conceivable that the thalamic explanation simply suffices to explain transmodal plasticity and that thalamic changes alone could explain previous observations (which would be an exciting conclusion to think about). That would make some good sense because the retino-LGN connection is probably underdeveloped in the CB child, hence the somatosensory fibers innervate the vacated LGN as a first step and any cortical organization follows. In other words, it is this fundamental earliest reorganization which is the exciting new understanding as it suggests a new cause-effect chain: the cortical (re-)organization is not the cause but the consequence of cross-modal plasticity. It is up to the authors to decide if such a bold conclusion can be made, but if sustainable and compatible with existing reports by others, it would be a daring but justified claim. If the authors follow this argument, then this should also be reflected in the Conclusions (P11)

Sur and Frost are certainly good references for examples of developmental plasticity, but the work of G.E. Schneider deserves to be quoted here. His work in hamsters was perhaps among the first to show re-routing of neural connections after early visual system lesions.

It might be helpful to explain how such studies of brain network reorganization are relevant for clinical "translation" by building bridges to adult vision loss such as glaucoma or PCA stroke? There are recent papers of brain network reorganization after visual system lesions (e.g. Bola et al. 2014, Neurology or Sabbah et al. 2017, Sci. Reports).

P10:a more direct demonstration of..

P11: Participants-section: Please state if any (or how many) of the CB subjects had any vision at all. If so, explain level of vision. Furthermore, specify how long the normal subjects were blindfolded (just during the recording or for extended times before recording?)

P16: Functional connectivity analysis. The alpha-range is defined as 8-12 Hz and the frequency bands of 13-14 are missing. Why? There are two sub-bands of alphas 8-11 (alpha I) and 12-14 (alpha II). Have the authors looked at them as well. Nevertheless, the results are pretty clear as is, so this issue is not critical. Along those lines: how about other frequency bands: delta, theta, and 30-40 Hz?

P23: The y-axes should be relabelled. All 3 graphs contain "mean" values, not only Fig. 1c. It would make sense to use the terms of the y-axis as a short title for each sub-figure and then specify the units to the y-axis. It would also be good if the authors state which areas of the brain are involved by adding that to the legend and by adding abbreviations to the graphs (e.g. Brodman numbers or names)

P25: This is a nice graph, but the lines and letterings are too thin/ too small. They will disappear in production. Also, the graph indicates "no eye". That is probably not the case. The subjects do have eyes (I presume), but their optic nerve/-path is damaged. Another graphic representation of that is needed. Simplify the graph by removing the brown shading of the brain. It serves no purpose

P28: specify the denomination of RIOS

P29: This reviewer would prefer that all individual responses are shown: One graph with the CB subjects and one for controls. It could be arranged in a way that each individual curve can be seen (similar to graphs of Fourier Analyses)

P30: lettering of these graphs are too small

Reviewer: B.A. Sabel

Response to reviewers

We are grateful for the time the reviewers spent on reading and commenting the first version of the manuscript. We would like to thank them for their many insightful remarks and questions. We have addressed all of their comments, also adding new analyses and new data that further strengthen our earlier conclusions. We are of the opinion that the revised version of the manuscript has gained a lot in quality. Below we respond on a point-by-point basis to the comments raised by the three reviewers.

Reviewer #1:

The authors present novel electrophysiological evidence on thalamocortical rerouting in congenitally blind subjects related to tactile processing. The methods used in the study are relevant and wisely selected. Statistical approaches used are valid. The manuscript is well written and easy to follow.

The manuscript has some minor issues which are listed below:

- Blindfolded CS is stated in the manuscript that the methods part does not describe how the CS subjects were blindfolded.

We have now added to the methods section details on the blindfolding during the recording sessions. See also Response to Reviewer 3 with respect to the same question.

- The way how range is expressed in the manuscript varies, e.g., 15-17 ms and 5 – 15 ms. The authors should use one common way. I personally would prefer 15–17 ms format.

Thanks for the observation. We have now standardized the format for expressing range to the one suggested by the reviewer (15–17 ms format).

- Page 2, Abstract: The authors use the term “blind brain” (abstract and discussion page 6). I guess that it would be better to use a term “brain in congenitally blind subjects” or similar.
Line 4 MEG = magnetoencephalography (MEG)

Thanks, the term has been updated and MEG defined.

- Line 7 The abbreviation LGN has not been defined. The abbreviation V1 has not been defined.

Thanks again for the observation. Both terms have now been defined.

- Line 10 blind participants => congenitally blind subjects

Thanks, the term has been updated.

- Page 3, Introduction: Line 22 The abbreviation MEG has not been defined.

Thanks, the term MEG has now been defined.

- Page 4, Results Behavioural data: Line 1 The authors should indicate what kind of stimuli were used and where they were applied in the first sentence of the results section. Now the first sentence is unclear.

A suggestion: The behavioural results indicate that supra-threshold electrocutaneous stimulation applied on the left index finger resulted in faster reaction times in CB compared with SC subjects.

Thanks for the suggestion. This has now been updated.

- How about the right index finger stimulation? Any statistics available?

The behavioral results for right index finger were not significant (CB: 392 ± 74 ms; SC: 376 ± 36 ms; $P > 0.05$). We have now added the results for right finger stimulation to the manuscript (see also response to reviewer 2).

Line 2 The abbreviation SC has not been defined.

Thanks, this abbreviation has now been defined.

- Page 4, Results MEG data: Line 1 The abbreviation SI has not been defined.

Thanks, this abbreviation has now also been defined.

- Line 7 Clumsy sentence: "In CB (but not in blindfolded sighted) participants, we identified significantly increased activity above pre-stimulus baseline ($p < 0.019$) in the contralateral occipital cortex as early as 35 ms following left index finger stimulation (Fig. 1b)." Rephrase it, e.g., "We identified significantly increased activity above pre-stimulus baseline ($p < 0.019$) in the contralateral occipital cortex as early as 35 ms following left index finger stimulation (Fig. 1b) only in CB subjects."

Thanks for the suggestion, the sentence has now been rephrased.

- Page 5, Line 4 The abbreviation V1 has not been defined. Line 13 The abbreviation FDR has not been defined.

Thanks, it has now been defined.

- Page 6, Line 1 The abbreviation SII has not been defined. Line 2 The abbreviations MT, V3, and PO have not been defined.

Thanks, they have now been defined.

- Line 13 "Alpha oscillations are prominently expressed over the posterior aspects of the occipital and parietal cortices in humans.²³ Is the citation correct?"

Yes, this statement is correct. Work from our group (Niso et al., Neuroimage 2016; 124, 1182-7) shows indeed that alpha oscillations in resting-state MEG are prominently expressed over the posterior aspect of the occipital and parietal cortices. See the image below:

- Line 25 "... blind brain." See the comment above (Page 2 abstract)
Thanks, this has also been updated.
- Page 7, Line 2 beta frequency range has not been defined. Line 16 The abbreviation PTE has not been defined.
Thanks, beta range has been included. PTE was defined earlier on page 9.
- Page 10, Line 16 FDG-PET has not been defined
Thanks, this has been defined.
- Page 13, Line 8 the brand and make of the response pads is missing.
This has now been added to the manuscript (see section Materials and Methods → Task)
- Page 14, Line 9 Polhemus Fastrack – country of the manufacturer is missing.
We have added the country of the manufacturer (Vermont, USA)
- Page 24, Figure 2 TH, V1, S1 have not been defined in the figure nor in the figure legend. Control and Blind are used although CS and CB are being used elsewhere
Thanks again for this careful observation. We have now used standard terminology for all the figure legends.
- References:
#23 is the citation correct in the manuscript? It does not state anything about parietal alpha activity.
It is the correct reference Niso G et al. (NeuroImage 2016). As shown in Figure 3 of that paper, the area of averaged relative power spectral density in the alpha band extends from occipital into posterior parietal areas.
#17 and #18 have different format with the year compared with the others.
Thanks, that has been corrected.
- Figure 1. The colors indicating standard deviations are not clearly visible, especially the yellow shade.
The figure has been redrawn and should be much clearer now.
- Supplementary Figure 2. Since there are no space limitations here, I would prefer to see the responses from thalamic region in each CB and CS subjects.
We have now added the thalamic group response in the new Figure 2, so there seems to be less need for showing all the individual thalamic responses.

- Supplementary Figure 3. How many averages were used for event-related time series?

This figure has now been removed.

Reviewer #2

This manuscript proposes that congenital blindness leads to cross modal reorganization of thalamic projections to V1 relaying somatosensory information to primary visual cortex. The conclusion is based on MEG data from 8 congenitally blind (CB) and 8 sighted controls (SC) in terms of source activity in time (relative to the sensory event) and brain anatomy. There is also functional connectivity data assessing strength of functional coupling across Thalamus, V1 and S1. The data contribute to an important and longstanding question regarding the where and how of cross modal reorganization after visual sensory deprivation in humans, the conclusions are novel and of relevance to the wider scientific community. As the authors summarize appropriately in their introduction to date work in human participants supports re-organization along cortico-cortical routes, whilst not directly supporting or ruling out reorganization via routes such as thalamus due to limitations in the method. The MEG method used here has high temporal resolution and ability (via source reconstruction) to link activity to certain neural structures even deeper in the brain, and both of these features make it a suitable tool to get at this question in humans. It is my opinion that such a finding is important and would influence thinking in the field. Yet, in its present form it is my opinion that the data as reported do not support the conclusions drawn. My concerns can be addressed through clarification in writing and by a suitable control experiment or additional data analysis.

My major concerns are (1) lack of reporting of relevant results in response to right finger stimulation (2) lack of sufficient separation of early responses (possibly thalamic projection driven) from later responses (possibly cortically projection driven) in V1 response average and spatial maps (3) lack of a visual control task in sighted control subjects (4) lack of reporting of group data for thalamic responses.

We thank the reviewer for the overall positive evaluations and for his/her insightful remarks about issues that in his/her opinion were not sufficiently convincing. We will start by addressing the four major concerns on a point by point basis:

(1) lack of reporting of relevant results in response to right finger stimulation. Reading through the manuscript it felt as if all results were only reported for left finger stimulation, but not right finger. I was not sure if no result was found for the right finger or if it was just not reported? If the results did not replicate with the right finger it is a reason for concern, unless you could explain why thalamic projections would reorganize differently for the two hands.

The reviewer was right that we only reported responses to left finger stimulation in the first version. In the current revised version of the manuscript we now present also the data of right hand stimulation in Supplementary Figure 2, and a new linear regression model of ROI

activations from pooled left and right hand stimulation trials, with augmented statistical power and confirmation of our previous results using a different approach.

From using the original distributed source model, we did not find a significant early V1 response to right-hand finger stimulation, possibly due to limited statistical power caused by relatively moderate number of subjects imposed by the CB special population. However, the new results from the linear regression model confirmed V1 activation in the CB group initiated as early as 35 ms and deviating significantly from the SC data.

(2) lack of sufficient separation of early responses (possibly thalamic projection driven) from later responses (possibly cortically projection driven) in V1 response average and spatial maps. The V1 responses driven by thalamic projections are expected to occur around 35ms latency. In fact, in the discussion you outline that the late component around 55ms is more likely to reflect contribution of cortical projections to V1. Yet, for spatial maps (Figure 1d) and averages (Figure 1c) you average from 30 to 55ms. Looking at Figure 1b suggests that this will then include the second peak at ~55ms, which seems separate from the first peak at about 35ms, and which is certainly too late to provide strong support for thalamic projections. As a consequence I am not sure that the spatial maps and averages you show are indeed linked to the early response as you state, or if they rather reflect the later responses in which case they are not supporting thalamic projection changes. I suggest redoing these analyses using instead an integration window of 20-35ms or 20-40ms. It might also be useful to compute averages and maps sequentially for 20 or 25ms bins shifting them by 1ms each and visualize them as they change over time. If there is indeed a thalamic projection averages in V1 and spatial maps w/r to V1 should diverge between blind and sighted from 30-35ms.

We thank the Reviewer for this comment and suggestion, which hopefully have contributed to strengthen the analysis and interpretation of our data. We have indeed performed a second analysis of the MEG data based on a linear regression model restricted to the three ROIs. We used the approach proposed by Coffey et al.¹⁷ to discriminate between the respective contributions from each brain region of interest to the MEG data. This linear mixing model has enhanced spatial specificity but is strictly limited to the 3 tested ROIs in comparison to the distributed model of neurophysiological responses presented above (see revised Methods). Following the Reviewer's recommendations, we applied temporal smoothing of the regression coefficients over a 15-ms sliding time window centered at each time point.

Akin to Coffey et al., who clarified cortical vs. deep and early brainstem responses to auditory stimulation using this approach, the time series of the linear regression model coefficients showed a distinct peak around 20 ms in Thal following finger stimulation in both groups (Fig. 2a). Similar to the distributed source data, the model did not show significant differences in Thal

and S1 between groups. It also confirmed the early V1 response in CB participants initiated as early as 35 ms after finger stimulation, with strong differential intensity compared to SC controls until at least 74 ms post-stimulus (Fig. 2b).

We emphasize that the 15-ms smoothing time window used in the linear mixing model was centered on each time point. Hence the early 35-ms onset was contributed by V1 signals within at most the 28-42 ms time range and cannot be explained by the secondary, later activity shown by the fully distributed cortical model (Fig 1b). Note that this latter did not include moving-average temporal smoothing, and indicated an early V1 peak in the CB group around 35 ms which was absent in SC. Taken together, we believe the new analyses underscore the presence of an early occipital peak in our CB subjects, and further strengthen our conclusions that we had put forward in the first version.

(3) Lack of a visual control task in sighted control subjects that would strengthen argument. As I perceive it, the main support in favour of the conclusion that there is cross-modal reorganization in thalamic projections towards V1 is based on the speed of the V1 response in blind participants (Figure 1b) and differences in spatial distribution of responses between blind and sighted participants (Figure 1c). W/r to those see also my previous comment. Whilst I appreciate the data as shown w/r to S1 (Figure 1a) showing that the two groups do not differ in terms of time courses in S1, using the same methods etc with a visual response in V1 in sighted controls would strengthen the conclusion that CB participants do indeed use 'visual' thalamic projections for relaying somatosensory information to V1. This is also the case because standard deviations around the mean in V1 at that early time point (35ms) are large and overlap between CB and SC participants (as far as I can tell based on Figure 1b), thus visual control data would show how this compares to visual response in SC. With respect to the functional connectivity analysis (PTE) I was surprised about the results in the beta band, where sighted show more connectivity from V1 and S1 to Thalamus as compared to blind. Whilst I appreciate the interpretation of this finding in the discussion, it would be good to demonstrate that the method will recover coupling in a visual task in sighted controls. This would strengthen the argument.

We thank the Reviewer for this insightful suggestion. We did not have a visual control experiment in our original study. However, we have now added data from a publicly available dataset consisting of a visual presentation task in a sighted subject (Supplementary Figure 2).

The selected data consisted of the visual presentation in 96 trials of faces to a participant seated under a similar Elekta MEG instrument as the one used to collect our own data. We defined the same ROIs as for the SC and CB groups. The resulting traces show an early response from the posterior aspect of the thalamus (Thal) following visual presentation (0 ms), which preceded the response peak in primary visual cortex (V1). Shading indicates the standard error on the mean.

We observed similar effects in terms of timing for LGN (Thal) and V1 after 18 and 36 ms, respectively.

Citation:

Wakeman DG, and Henson RN. 2015. "A Multi-Subject, Multi-Modal Human Neuroimaging Dataset." *Scientific Data* 2 (January): 150001. <https://doi.org/10.1038/sdata.2015.1>.

(4) Lack of reporting of group data for thalamic Responses. Two examples of thalamic responses are provided in the supplemental materials. Yet, it is my opinion that if you argue that your method reliably measures thalamic responses (as needed for your PTE analysis) I strongly suggest including group data for those responses.

We thank you for this comment. We have now included the results of the new analysis reflecting the timing of each region's contribution for both groups which illustrate a peak activity at 20 ms in both groups (Fig. 2a).

Minor concerns

(5) lack of consideration of effects of Braille experience/reliance on touch vs. blindness. You do not report details on participants, incl. usage of Braille or touch. There is the possibility that if all blind participants are proficient and regular users of Braille the reorganization is in response to prolonged use of tactile information and not blindness. These aspects of blindness (i.e. lack of visual input vs stronger reliance on other modalities such as touch more generally or Braille specifically) are difficult to disentangle. You also show that the people who are blind in your study had lower threshold levels, suggesting heightened sensitivity for somatosensory stimulation. Thus it would be good to discuss this aspect w/r to your interpretation.

We thank the reviewer for raising this interesting comment. All our CB subjects were proficient Braille readers which raises indeed the possibility that some of the changes may relate not to blindness per se but to the fact that our subjects rely more heavily on tactile input. There is indeed some evidence from published literature that relying more strongly on the tactile channel may trigger cross-modal responses. For instance, work by the group of Amedi has shown that extended Braille training in sighted subjects may trigger cross-modal responses in the occipital cortex in response to Braille reading, although this activation seemed limited to extrastriate visual areas (Siuda-Krzywicka et al., 2016). In blind subjects, Wong and co-workers showed that reliance on the sense of touch, and not blindness per se, is the trigger for tactile spatial acuity enhancement.

In further support of a role of increased practice (rather than blindness per se) is the observation by Voss and Zatorre (2013) that anatomical changes (cortical thickness) in occipital areas are directly related to heightened behavioral abilities in blind subjects. Since all of our congenitally blind subjects were proficient Braille readers, it is difficult to dissociate whether the early occipital activation is due to blindness per se or to the increased reliance on the tactile modality. We have added the following text to the discussion:

“Since all our CB subjects were Braille readers, another limiting factor is that some of the observed changes may not relate to blindness per se but to the fact that our subjects rely more heavily on tactile input. There is indeed evidence from the literature that increased reliance on the sense of touch is the trigger for both behavioural and brain functional and structural changes (Wong et al., 2011; Vos et al., 2012; Siuda-Krzywicka et al., 2016)”

Refs:

Siuda-Krzywicka K, Bola Ł, Paplińska M, Sumera E, Jednoróg K, Marchewka A, Śliwińska MW, Amedi A, Szwed M. Massive cortical reorganization in sighted Braille readers. *Elife*. 2016 Mar 15;5:e10762. doi: 10.7554/eLife.10762.

Voss P, Zatorre RJ. Occipital cortical thickness predicts performance on pitch and musical tasks in blind individuals. *Cereb Cortex*. 2012;22:2455-65.

Wong M, Gnanakumaran V, Goldreich D. Tactile spatial acuity enhancement in blindness: evidence for experience-dependent mechanisms. *J Neurosci*. 2011 May 11;31(19):7028-37.

(6) Lack of detail of blind participant characteristics, i.e. level of remaining vision/ light perception, use of Braille. See also my comment (5)

We have added a table (Supplementary Table 1) with the demographic table on the blind participants.

(7) Figure 1 c and d have been reversed with respect to the caption.

Thanks, this figure has been updated.

(8) Previous work has shown differences in alpha band activity in blind compared to sighted people (Kriegseis et al. Reduced EEG alpha activity over parieto-occipital brain areas in congenitally blind adults. *Clinical neurophysiology* 117, no. 7 (2006): 1560-1573) which might be worth mentioning in the context of your results re connectivity in alpha bands.

We have added the following text to the discussion:

“We note that Kriegseis and co-workers³⁰ reported a reduced EEG alpha activity over parieto-occipital brain areas in CB in a somatosensory perception and mental imagery task. This finding is not in contradiction with our data, which report effects in alpha inter-regional directed connectivity rather than differences in power. Further, we report occipital activity from the primary visual cortex rather than from an extended parieto-occipital region.”

(9) It was not entirely clear to me which trials were entered into the analyses. You have Comfort level and supra level trials and attended and non-attended. Which trials were used or were they all combined? Please clarify.

Thanks for this remark. We have clarified this issue now and added the following to the manuscript (Method section):

“Only the results of the supra-threshold stimulation in the attended condition are shown in this paper.”

(10) Please specify which nuclei your ROI ‘posterior thalamus’ contains.

We have now specified that the posterior thalamus ROI included the ventroposterior lateral thalamus (VPL) and the lateral geniculate nucleus (LGN).

(11) I was wondering if you had also considered contributions and cross modal reorganizations from Superior (or possibly even inferior) colliculus and relevance to the responses you measured in V1. Animal work suggests that these nuclei and their projections show structural and functional changes in response to sensory/vision loss.

This is an interesting suggestion as the rerouting of tactile information to the visual cortex could theoretically also be mediated by the superior colliculus (SC). However, a rerouting of tactile information via the superior colliculus would be strongly polysynaptic (from thalamus to SI, superior colliculus, pulvinar and then to V1) and is incompatible with our measured response latencies. In case we had used auditory instead of electro-tactile stimulation, a set of ectopic connections could explain visual cortex activation by auditory stimulation: between the inferior colliculus and the dLGN, V1 and the superior colliculus, the lateral posterior nucleus and V1 and lastly the superior colliculus and the medial geniculate nucleus.

Reviewer #3

General comments regarding the manuscript: Are they novel and will they be of interest to others in the community and the wider field? Yes, they are of general interest to a wider field. Is the work convincing? Yes. On a more subjective note, do you feel that the paper will influence thinking in the field? Absolutely. We would also be grateful if you could comment on the appropriateness and validity of any statistical analysis, as well the ability of a researcher to reproduce the work, given the level of detail provided: Yes, statistics are appropriate and the work can be reproduced by others.

Specific comments regarding the manuscript: People with blindness since birth (congenital blindness, CB) have particularly well developed other senses like hearing and touch. The biological basis of these “super-normal” abilities is an altered brain developmental organization (not re-organization) whereby cortical regions normally used for visual processing (visual cortex) are “invaded” by neuronal connections from the other sensory system. It is generally thought that this “cross-modal plasticity” involves the alteration of cortical connectivity, whereby, for example, tactile stimulation is rerouted from the primary somatosensory cortex via different cortical connections to the occipital cortical regions where (in normal subjects) vision would have resided.

The authors have now shown for the first time that such cross-modal plasticity involves a much more fundamental brain connectivity change whereby the reorganization is already taking place in the subcortical (thalamic) region, i.e. the first relay nucleus where the different sensory systems meet. In other words, it is this fundamental first step that could explain cross-modal plasticity, long before information arrives at the cortical level. According to their paradigmatically new proposal, it is the short VPL-to-LGN reorganization which could explain cross-modal plasticity. The key experimental observation by the authors was that occipital cortical activation at 35 ms following tactile stimulation of the index finger using MEG recordings. In contrast to previous PET and fMRI studies, this technique had an excellent temporal resolution which allowed the measurement of such fast changes.

This new discovery is of significant public interest as it not only informs us of the biological basis of the enormous brain plasticity potential of the human brain but in that it provides a convincing explanation of a long-sought answer as to why people with early vision loss have improved (“super-normal”) performance in other sensory system. The study was well designed, the techniques used are up-to-date and the data analysis appropriate, the conclusions are well grounded in the data. There is no doubt that this paper is a highlight in the field of cross-modal plasticity and deserves publication in Nature Communications. It is a discovery that will draw attention not only by specialists in the field but also by the public at large.

We thank the Reviewer the very positive opinion about our study and the manuscript.

Minor comments worth considering for revising the paper.

Page 2: Abstract: The first sentence should include the info “...congenital blindness” or “...human brain blind since birth” or similar.

Thanks, this has been updated.

The authors should state also clearly, that they looked at a case of developmental plasticity, i.e. a modification of brain organization so that readers do not confuse this with adult post-lesion plasticity or reorganization (this comment also applies to the introduction and discussion). The abstract does not do justice to the fundamental discovery of the work. The rationale for this experiment as stated is that “...the contribution of a sub-cortical pathway cannot be excluded”. The argument that something can not be excluded is not an argument that raises excitement – it is underselling the significance of the paper. The “helicopter” description above may help the authors to come up with a better (more substantive and sophisticated) argument why the study was carried out and why it solved a milestone problem of brain developmental plasticity.

We have changed the abstract now accordingly. The rationale for the study has now been rephrased as follows:

“Most studies on the developmental sensory reorganization of the brain of congenitally blind individuals suggest that cross-modal responses are mediated by a strengthening of existing cortico-cortical pathways. However, some animal studies raise the possibility that a subcortical pathway could also be involved (Bronchti et al., 1989; Karlen et al., 2006) which would allow fast re-routing of tactile information to the occipital cortex. To resolve this controversy, we combined functional activation and directed connectivity magnetoencephalography (MEG) analyses in human subjects.”

Add “...blindfolded sighted control subjects”...

Thanks, this has been added.

P 3 Introduction: Again, the authors should point out that developmental plasticity is different from adult post-lesion plasticity. In other words, there may be many similarities between both, but during development fiber connections are just beginning to be formed (organization), whereas in adulthood lesions/damage, they are already in place. Thus, while such studies with congenitally blind can give us hints for any conclusions about plasticity of the human brain in adulthood, developmental plasticity is a different matter. That is a limitation which deserves to be mentioned.

We have now added to the introduction that our data only relate to congenital blindness or developmental plasticity. We have added the following text to the introduction:

“This condition needs to be clearly distinguished from adult post-lesional plasticity, or late-onset blindness, whereby blindness occurs at a moment the wiring of the brain has already been firmly established.”

P4 Results: To facilitate readability, the first sentence in this section should mention that the authors compared CB participants with normally sighted subjects which were blindfolded for xxx min/hrs.

Thanks, this has been added (see section Materials and Methods → Participants).

P5 Discussion: The discussion should start off with a short “helicopter” view of the study rationale, then state the key finding again.

Thanks, this has been added to the first paragraph of the discussion.

P8: Last sentence of 2nd paragraph (...“Our results also provide...”). Here it would be helpful if the authors state if the thalamic reorganization coexists as a “second” mechanism of transmodal plasticity or if it “replaces” the concept of cortico-cortical reorganization. In other words, is it conceivable that the thalamic explanation simply suffices to explain transmodal plasticity and that thalamic changes alone could explain previous observations (which would be an exciting conclusion to think about). That would make some good sense because the retino-LGN connection is probably underdeveloped in the CB child, hence the somatosensory fibers innervate the vacated LGN as a first step and any cortical organization follows. In other words, it is this fundamental earliest reorganization which is the exciting new understanding as it suggests a new cause-effect chain: the cortical (re-?)organization is not the cause but the consequence of cross-modal plasticity. It is up to the authors to decide if such a bold conclusion can be made, but if sustainable and compatible with existing reports by others, it would be a daring but justified claim. If the authors follow this argument, then this should also be reflected in the Conclusions (P11) Sur and Frost are certainly good references for examples of developmental plasticity, but the work of G.E. Schneider deserves to be quoted here. His work in hamsters was perhaps among the first to show re-routing of neural connections after early visual system lesions. It might be helpful to explain how such studies of brain network reorganization are relevant for clinical “translation” by building bridges to adult vision loss such as glaucoma or PCA stroke? There are recent papers of brain network reorganization after visual system lesions (e.g. Bola et al. 2014, Neurology or Sabbah et al. 2017, Sci. Reports).

We thank the reviewer for this thoughtful and provocative viewpoint. We have replied to his comment by adding the following text to the Discussion:

“This raises the question of the relative contributions of the thalamocortical and cortico-cortical pathways in cross-modal plasticity. Phrased differently, would the thalamo-cortical pathway be sufficient to explain by itself cross-modal responses? Although we cannot give a conclusive answer to this question, based solely on the present findings, we believe it is unlikely that of the thalamo-cortical connection would be sufficient to fully explain cross-modal plasticity. First, the magnitude of the thalamo-cortical response is quite moderate compared to the later cortico-cortical response. Second, there is evidence that blocking the cortico-cortical pathway by applying TMS over S1 interferes with cross-modal plastic responses (Wittenberg et al., 2004). We suggest that the arrangement of a dual pathway to the occipital cortex in the congenitally blind brain allows early thalamo-cortical signalling to modulate ensuing occipital responses communicated by the slower cortico-cortical pathways, facilitating re-allocation of visual cortex to tactile processing.”

P10:a more direct demonstration of...

Thanks, this has been updated.

P11: Participants-section: Please state if any (or how many) of the CB subjects had any vision at all. If so, explain level of vision. Furthermore, specify how long the normal subjects were blindfolded (just during the recording or for extended times before recording?)

Thanks, this has been added. None of the participants reported any remaining vision.

P16: Functional connectivity analysis. The alpha-range is defined as 8-12 Hz and the frequency bands of 13-14 are missing. Why? There are two sub-bands of alpha 8-11 (alpha I) and 12-14 (alpha II). Have the authors looked at them as well. Nevertheless, the results are pretty clear as is, so this issue is not critical. Along those lines: how about other frequency bands: delta, theta, and 30-40 Hz?

We thank the reviewer for his remark on the frequency bands for the connectivity analysis. We also looked at some other frequency domain and we found that there were also some significant changes in functional connectivity in the beta band range, but not in delta, theta or gamma band range. More specifically, sighted controls had stronger connectivity between V1 and Thalamus and between S1 and Thalamus. These new data have now been added to the manuscript and Figure 3 has been updated to incorporate these new findings.

P23: The y-axes should be relabelled. All 3 graphs contain “mean” values, not only Fig. 1c. It would make sense to use the terms of the y-axes as a short titles for each sub-figure and then specify the units to the y-axis. It would also be good if the authors state which areas of the brain are involved by adding that to the legend and by adding abbreviations to the graphs (e.g Brodman numbers of names)

Thank you for suggestion, we modified the y-axis in the new version of Figure 1.

P25: This is a nice graph, but the lines and letterings are too thin/ too small. They will disappear in production. Also, the graph indicates “no eye”. That is probably not the case. The subjects do have eyes (I presume), but their optic nerve/-path is damaged. Another graphic representation of that is needed. Simplify the graph by removing the brown shading of the brain.

The figure has been adapted according to the stylistic suggestions by the reviewer (larger lettering, thicker lines and removal of the brown shading of the brain.

P28: specify the denomination of RIOS

The ROIs have been denominated now.

P29: This reviewer would prefer that all individual responses are shown: One graph with the CB subjects and one for controls. It could be arranged in a way that each individual curve can be seen (similar to graphs of Fourier Analyses)

We are of the opinion that showing the individual curves would create a rather messy figure. However, wherever we thought it was stylistically fine, we have added individual data points which provide a good estimate of the interindividual spread of the data.

P30: lettering of this graphs are too small

Thanks, we have now used bigger fonts in the new figures.

Reviewers' comments:

Reviewer #1 (Remarks to the Author):

The manuscripts has improved during the revision. All the questions and comments have been taken into account. Excellent work.

I do have one minor observation. The Figure 4 shows labels VIP and PO whereas the figure legend mentions PPC instead.

Reviewer #2 (Remarks to the Author):

I appreciate the inclusion of additional data and analyses provided, as well as modifications to the text, all of which improve the manuscript. Yet, I have remaining concerns.

In my initial review I had raised concerns regarding findings with right hand stimulation. The authors now provide data for results with right hand stimulation. When looking at the fully distributed model, there is no early response for the right hand in V1. The authors argue that moderate sample size might be an issue, but at the same time sample size is enough to yield a reliable effect for the fully distributed model for left hand stimulation. Thus, it is likely not sample size per se, but either sample size combined with heterogeneity of responses in particular in the left hemisphere, or just lack of response in the left hemisphere. In the previous round of reviews the other reviewer's had asked for individual data to be shown. This might help resolve this question. When right and left hands (and hemispheres) are combined in the multiple linear regression analysis there is an early response in V1. This result does help strengthening the argument. What I would suggest, however, to further strengthen the manuscript is to give more information on the differences between fully distributed model and multiple regression analysis. The way I understood it, time courses of the fully distributed model shown in Figure 1 and Supplementary Figure 3 are taking into account sensors in relevant ROIs based on Brodmann areas registered to the cortical surface, i.e. Thal, S1, V1, so you are showing time courses averaged across sources within those ROIs. The multiple linear regression analysis also uses ROIs, but assigns weights to different sensors within each ROI? If this is the case, then this is the reason for difference in results, but not the use of ROIs per se. I think it would help to give readers less familiar with the MLR method (like myself) a bit more information about what is actually happening. I do realize that the reader could also go on and read the paper by Coffey et al.,

but since the results are important for the arguments made here, it might be helpful to give a bit more of an intuition.

Related, in Figure 2 you show results from the multiple linear regression analysis. In the bottom panel (Figure 2b) you compare beta weights for [35-50ms] in V1 for SC and CB. Since there are 8 participants in each group, I was expecting to see 8 data points in each group, one for each participant. Yet, there are 16 data points for each group. Is this because you put in beta weights for left and right separately? In this case, however, I am not sure how you computed the statistical test. Specifically, to compare data from SC and CB I would assume that you could use an independent samples t-test, but if you are also including left and right from the same participants, this would not be correct. I suppose in that case you would have to use a mixed model ANOVA, i.e. hemisphere (left, right) as within subject variable, and group (SC, CB) as between subject variable? Alternatively, you could average across left and right HS first, and then use an independent samples t-test. Either way, it would be helpful to have a bit more detail on what exactly data points in Figure 2b refer to. Also, in Figure 2 caption referring to the three panels in Figure 2a you state "Grey shaded boxes highlight segments of significant differences between the two groups ($p = .038$)...". Is this the critical p-value at which you evaluate significance? In this case, why not use $p < .05$? Also, was significance determined for each time point separately? Which test was used, i.e. did you average across left and right HS first then do independent samples t-test to compare SC and CB?

In my original review I had pointed out a possible issue with lack of sufficient separation of early responses (possibly thalamic projection driven) from later responses (possibly cortically projection driven) in V1 response average and spatial maps (i.e. data shown in Figure 1 and supplementary Figure 3). To address my concern the authors have used a different time integration window for the multiple regression analysis. Yet, my concern still applies to the fully distributed model and analysis, and as a consequence I am still wondering to what degree spatial maps shown in Figure 1 are driven by later cortical responses as opposed to earlier ones. I realize that I am being picky here, but since the main argument of this manuscript is about the early response in V1, it still is my opinion that these early responses (and spatial maps related to them) should be separated from later responses. Please see my previous review for more detailed suggestions. Maybe I am missing something here, but in this case please clarify.

Below I outline some other concerns as they arise in the manuscript.

Introduction page 5, and also later in text: you refer to Figure 1 panels a-c, but Figure 1 only has panels a and b. Please make consistent.

Introduction page 5, and also later in text: when you refer to results from right index Finger you refer to Suppl. Figure 2, but this is actually supplementary figure 3. Please make consistent.

Stats reporting: you typically only report p-values; it might be useful to report test statistics as appropriate and to also state if corrections for multiple comparison were used (you state this for functional connectivity analyses but not for the other tests).

You conducted a control analysis using a data set from a single participant in a visual task, but you actually do not refer to this analysis or result in your main manuscript text so that the reader is never aware of it based on the manuscript text. I suggest referring to it at the appropriate point in the manuscript.

Reviewer #3 (Remarks to the Author):

All my suggestions for revision were adequately addressed and the paper is now publishable. The paper has improved significantly and makes a very convincing argument. The only minor editorial suggestions I would like to add are the following:

Abstract:

“...we identified occipital cortex activations following tactile stimulation, starting...

“in the cross-modal activation of the occipital cortex by tactile stimulation in congenitally blind humans”.

It might help the non-specialized reader to add some wording that occipital cortex does not normally respond to tactile input.

Discussion:

“...Since the LGN does not receive retinal inputs in CB individuals.... Please add reference here”

Response to reviewers' comments

Reviewer #1

The manuscripts has improved during the revision. All the questions and comments have been taken into account. Excellent work.

I do have one minor observation. The Figure 4 shows labels VIP and PO whereas the figure legend mentions PPC instead.

We thank the Reviewer for his positive comments on the new version of the manuscript. We have now corrected the legend of Figure 4 for consistency with the labels shown in the figure.

Reviewer #2

I appreciate the inclusion of additional data and analyses provided, as well as modifications to the text, all of which improve the manuscript. Yet, I have remaining concerns.

In my initial review I had raised concerns regarding findings with right hand stimulation. The authors now provide data for results with right hand stimulation. When looking at the fully distributed model, there is no early response for the right hand in V1. The authors argue that moderate sample size might be an issue, but at the same time sample size is enough to yield a reliable effect for the fully distributed model for left hand stimulation. Thus, it is likely not sample size per se, but either sample size combined with heterogeneity of responses in particular in the left hemisphere, or just lack of response in the left hemisphere. In the previous round of reviews the other reviewer's had asked for individual data to be shown. This might help resolve this question. When right and left hands (and hemispheres) are combined in the multiple linear regression analysis there is an early response in V1. This result does help strengthening the argument. What I would suggest, however, to further strengthen the manuscript is to give more information on the differences between fully distributed model and multiple regression analysis. The way I understood it, time courses of the fully distributed model shown in Figure 1 and Supplementary Figure 3 are taking into account sensors in relevant ROIs based on Brodmann areas registered to the cortical surface, i.e. Thal, S1, V1, so you are showing time courses averaged across sources within those ROIs. The multiple linear regression analysis also uses ROIs, but assigns weights to different sensors within each ROI? If this is the case, then this is the reason for difference in results, but not the use of ROIs per se. I think it would help to give readers less familiar with the MLR method (like myself) a bit more information about what is actually happening. I do realize that the reader could also go on and read the paper by Coffey et al., but since the results are important for the arguments made here, it might be helpful to give a bit more of an intuition.

We thank the Reviewer for this suggestion. His/her understanding of the similarities and differences between the fully distributed and the multiple linear regression model

is correct. In the present revised version, we have elaborated further on the distinction between the approaches in the Results section of the manuscript (pp. 5-6), which now reads as follows:

“We confirmed these findings with a second analysis of the MEG data based on a linear regression model restricted to the three ROIs. We replicated the approach used by Coffey et al. 17 to further discriminate between the respective contributions from each brain region of interest to the MEG data. As explained in the Methods section, the resulting multiple linear regression model (MLR) restricts the modelling of absolute-valued sensor MEG data to the instantaneous, weighted linear combination of the absolute values of the forward fields of each ROI. The ROI forward fields were obtained from the same head model coefficients as those used for the wMNE approach. The rectification of the sensor data and forward fields emphasizes the fit of each ROI’s contribution to the topography of recorded MEG sensor data, regardless of the flux direction of the magnetic induction measured outside the head. The resulting linear regression coefficients obtained for each region at every point in time therefore underlines each ROI’s contribution to surface data while lifting the ambiguity of estimating the direction of its current flow. This linear mixing model has enhanced spatial specificity but is strictly limited to the 3 tested ROIs in comparison to the distributed model of neurophysiological responses presented above (wMNE), which provided a fuller physiological account of cortical activations (i.e., by estimating both the amplitude and the direction of the current flow in all ROIs and the rest of the cortical surface; see Methods).“

Related, in Figure 2 you show results from the multiple linear regression analysis. In the bottom panel (Figure 2b) you compare beta weights for [35-50ms] in V1 for SC and CB. Since there are 8 participants in each group, I was expecting to see 8 data points in each group, one for each participant. Yet, there are 16 data points for each group. Is this because you put in beta weights for left and right separately? In this case, however, I am not sure how you computed the statistical test. Specifically, to compare data from SC and CB I would assume that you could use an independent samples t-test, but if you are also including left and right from the same participants, this would not be correct. I suppose in that case you would have to use a mixed model ANOVA, i.e. hemisphere (left, right) as within subject variable, and group (SC, CB) as between subject variable? Alternatively, you could average across left and right HS first, and then use an independent samples t-test. Either way, it would be helpful to have a bit more detail on what exactly data points in Figure 2b refer to.

Indeed, we pooled the data from both hemispheres to increase statistical power. We did not find a statistically significant main effect of hemispheric lateralization (not reported). Following the Reviewer’s suggestion, we now provide the results of a mixed-model ANOVA with hemisphere as “within-subjects” and group as “between-subjects” variable”. These results are now added to the revised version of the manuscript (p. 7):

“The effects of hemisphere and group were tested using a mixed model ANOVA with hemisphere as within-subjects variable and group as between-subjects variable. The results showed no main effect for hemisphere ($F(1) = 1.7, p = 0.21$) or interaction between hemisphere and group ($F(1) = 2.33, p = 0.15$), but there was a main effect

of groups ($F(1) = 15.6, p = 0.001$), reflecting stronger activity in V1 for CB subjects. (See Supplementary Table 4)”

Also, in Figure 2 caption referring to the three panels in Figure 2a you state “Grey shaded boxes highlight segments of significant differences between the two groups ($p = .038$)...”. Is this the critical p-value at which you evaluate significance? In this case, why not use $p < .05$? Also, was significance determined for each time point separately? Which test was used, i.e. did you average across left and right HS first then do independent samples t-test to compare SC and CB?

Thank you for pointing this out. This was a typo in the revised manuscript. As shown in the figure, the critical p-value used was indeed 0.05. As explained in the Methods and Discussion sections (pages 19 and 7, resp.), statistical significance was determined using a smoothing 15-ms window around each time point. We indeed used an independent two-sample two-tailed t-test on averaged data values between left and right homologous regions for the comparison between the two groups of participants.

In my original review I had pointed out a possible issue with lack of sufficient separation of early responses (possibly thalamic projection driven) from later responses (possibly cortically projection driven) in V1 response average and spatial maps (i.e. data shown in Figure 1 and supplementary Figure 3). To address my concern the authors have used a different time integration window for the multiple regression analysis. Yet, my concern still applies to the fully distributed model and analysis, and as a consequence I am still wondering to what degree spatial maps shown in Figure 1 are driven by later cortical responses as opposed to earlier ones. I realize that I am being picky here, but since the main argument of this manuscript is about the early response in V1, it still is my opinion that these early responses (and spatial maps related to them) should be separated from later responses. Please see my previous review for more detailed suggestions. Maybe I am missing something here, but in this case please clarify.

The Reviewer raises here indeed an important point. However, we believe we had already addressed this issue in the previous revised version of the manuscript. We kindly refer to page 11 (Discussion) where this specific topic was addressed:

“The 15-ms smoothing time window used in the linear mixing model was centered on each time point. Hence the early 35-ms onset was contributed by V1 signals at most within the 28-42 ms time range and cannot be explained by the secondary, later activity shown by the fully distributed cortical model (Fig 1b). Note that this latter did not include moving-average temporal smoothing, and indicated an early V1 peak in the CB group around 35 ms which was absent in SC.”

To avoid any further confusion, we now have also expanded the Results section (pp. 6-7) with:

“The MLR model was based on smoothed time series over a sliding 15-ms time window centered at each point in time. The resulting temporal smoothing involved data points ± 7 ms around the running data point at time t . For this reason, the early onset activity revealed in CB V1 at 35 ms may be contributed by regional activation

over the [28, 42]-ms time window across participants. Therefore, the later responses revealed by both the MLR and the wMNE model (Figs 1B and 2A) are not likely to have contributed to this earlier-onset V1 activation in CB, which was absent in SC. No time-windowing was used in the wMNE sources time series.”

Below I outline some other concerns as they arise in the manuscript.

Introduction page 5, and also later in text: you refer to Figure 1 panels a-c, but Figure 1 only has panels a and b. Please make consistent.

Introduction page 5, and also later in text: when you refer to results from right index Finger you refer to Suppl. Figure 2, but this is actually supplementary figure 3. Please make consistent.

These two points have been addressed: thank you.

Stats reporting: you typically only report p-values; it might be useful to report test statistics as appropriate and to also state if corrections for multiple comparison were used (you state this for functional connectivity analyses but not for the other tests).

Thank you for raising this point. In most cases, we had reported the type of statistical test used and the relevant parameters in addition to p-values. For instance, we reported t-values and the number of degrees of freedom. However, you rightfully point at some inconsistencies in the reporting, which we have now fixed in this newly revised version.

Legend Fig.2:

- Part a: we replaced ($p=0.038$) with *“two-tailed independent t-test, $p<0.05$ – not corrected for multiple comparison due to limited number of participants”*
- Part b: we replaced ($p=0.038$) with *“Mixed model ANOVA with hemisphere as within subject variable and group as between subject variable: no main effect of hemisphere or interactions between hemisphere and group, but significant effect of group ($F(1) = 15.6, p = 0.001$) reflecting stronger activity in V1 for CB subjects”.*

You conducted a control analysis using a data set from a single participant in a visual task, but you actually do not refer to this analysis or result in your main manuscript text so that the reader is never aware of it based on the manuscript text. I suggest referring to it at the appropriate point in the manuscript.

As requested by the Reviewer, we now refer to this control experiment in the main text of the manuscript (page 5).

Reviewer #3

All my suggestions for revision were adequately addressed and the paper is now publishable. The paper has improved significantly and makes a very convincing argument. The only minor editorial suggestions I would like to add are the following:

Abstract:

“...we identified occipital cortex activations following tactile stimulation, starting...

“in the cross-modal activation of the occipital cortex by tactile stimulation in congenitally blind humans”.

It might help the non-specialized reader to add some wording that occipital cortex does not normally respond to tactile input.

We have made these changes to the new version of the abstract: thank you.

Discussion:

“...Since the LGN does not receive retinal inputs in CB individuals.... Please add reference here”

A specific reference is now added to the revised version (Ptito et al., 2008): thank you.

REVIEWERS' COMMENTS:

Reviewer #2 (Remarks to the Author):

The authors have addressed my concerns.

My only remaining comment is that when reporting ANOVA results you should report df both for numerator and denominator. In your case this would be $F(1,14)$ for the mixed model ANOVA for all effects I think.